# Projections of Precipitation and Temperatures in Greenland and the Impact of Spatially Uniform Anomalies on the Evolution of the Ice Sheet

Nils Bochow[1], Anna Poltronieri[1], and Niklas Boers[2,3,4]

[1]Department of Mathematics & Statistics, Faculty of Science and Technology, UiT – The Arctic University of Norway, Tromsø, Norway.
[2]Potsdam Institute for Climate Impact Research, Potsdam, Germany.
[3]Earth System Modelling, School of Engineering & Design, Technical University of Munich, Germany.
[4]Department of Mathematics and Global Systems Institute, University of Exeter, UK.

**Correspondence:** Nils Bochow (nils.bochow@uit.no)

**Abstract.** Simulations of the Greenland ice sheet (GrIS) at millennial time scales and beyond often assume spatially and temporally uniform temperature anomalies and precipitation sensitivities over these time scales, or rely on simple parameterisation schemes for the precipitation rates. However, there is no a priori reason to expect spatially and temporally uniform sensitivities across the whole GrIS. Precipitation is frequently modelled to increase with the standard thermodynamic scaling of $\sim 7\%/$K derived from the Clausius-Clapeyron relation and often based on older model generations. Here, we update the commonly used parameters for long-term modelling of the GrIS, based on the output of the latest generation of coupled Earth system models (CMIP6), using the historical time period and four different future emission scenarios. We show that the precipitation sensitivities in Greenland have a strong spatial dependence, with values ranging from $-3\%/$K in southern Greenland to $13\%/$K in northeastern Greenland relative to the local annual mean near-surface temperature in the CMIP6 ensemble mean. Additionally, we show that the annual mean temperatures in Greenland increase between $1.29$ and $1.53$ times faster than the global mean temperature (GMT), with northern Greenland warming up to two times faster than southern Greenland in all emission scenarios. However, we also show that there is a considerable spread in the model responses that can, at least partially, be attributed to differences in the Atlantic meridional overturning circulation (AMOC) response across models. Finally, using the Parallel Ice Sheet Model (PISM), we show that assuming uniform temperature and precipitation anomalies and sensitivities leads to overestimation of near-surface temperatures and underestimation of precipitation in key regions of the GrIS, such as southwestern Greenland. This, in turn, can result in substantial overestimation of ice loss in the long-term evolution of the GrIS.

## 1  Introduction

The Greenland ice sheet (GrIS) is the second largest terrestrial ice sheet with the potential of more than $7.4\,\mathrm{m}$ of sea level rise when completely melted (Morlighem et al., 2017). Increasing atmospheric and oceanic temperatures due to climate change (Straneo and Heimbach, 2013; Fettweis et al., 2017) led to more than a 5-fold increase in ice loss from the Greenland ice

sheet in the last three decades (Shepherd et al., 2020; Otosaka et al., 2023). Land ice melt contributed more than half of the global sea level rise since 1993 with an acceleration in recent years (IPCC, 2022). The total mass balance of the GrIS, i.e. the difference between total mass gain and loss, has been decreasing steadily in the last decades, with an average yearly mass loss of $169 \pm 9$ Gt/yr between 1992 and 2020 (Otosaka et al., 2023). This has caused a cumulative global sea level rise of more than 12 mm (Shepherd et al., 2020; Otosaka et al., 2023). Alternatively, the total mass balance can be seen as the sum of the surface mass balance (SMB), the discharge into the ocean (D) and the basal mass balance (BMB). The decrease in total mass balance of the GrIS in the last decades is divided approximately equally between discharge due to ice dynamics (D) and increased surface melt, i.e., decreasing SMB (van den Broeke et al., 2009; Choi et al., 2021). However, projections show that increases in Greenland surface melt will dominate the decrease in the mass balance in the long-term, mostly due to a retreating ice front and the diminishing contact of the ice sheet with the ocean (Goelzer et al., 2020; Choi et al., 2021; Payne et al., 2021).

Many studies have investigated the past and future evolution of the GrIS from short to long timescales, using computational methods ranging from simple conceptual models (Levermann and Winkelmann, 2016; Boers and Rypdal, 2021), stand-alone ice sheet models (Bochow et al., 2023; Goelzer et al., 2020; Seroussi et al., 2020) to Earth system models (ESMs) of intermediate complexity and full complexity with dynamically coupled ice sheets (Robinson et al., 2012; Gregory et al., 2020; Höning et al., 2023, 2024; Muntjewerf et al., 2020a, b; Madsen et al., 2022). In the latest generation of comprehensive ESMs from the 6th phase of the coupled model intercomparison project (CMIP6) (Eyring et al., 2016), the ice sheets were not fully bidirectionally coupled to the other components. However, dynamically coupled ice sheets have successfully been introduced recently in several ESMs (Mikolajewicz et al., 2007; Vizcaíno et al., 2008; Muntjewerf et al., 2020b; Smith et al., 2021). While this coupling is making rapid progress, parameterisation schemes are still needed, especially to investigate the long-term behaviour of ice sheets. Computational constrains make it currently challenging to run ESMs on millennial or even decamillennial timescales.

Long-term simulations of the ice sheet usually rely on parameterisation methods, for instance, for calculating the SMB. Commonly used methods include positive-degree day methods (PDD) (Huybrechts and Oerlemans, 1990) or surface energy balance models (Krebs-Kanzow et al., 2021). These models calculate the SMB based on several inputs such as near-surface temperature and precipitation fields. Two of the most commonly used approaches to produce these forcing fields for long-term ice sheet modelling are (1) the use of a fixed precipitation increase per degree of temperature increase (e.g., Greve et al., 2011; Pollard and DeConto, 2012; Saito et al., 2016; DeConto and Pollard, 2016; Cuzzone et al., 2019; Aschwanden et al., 2019; Rodehacke et al., 2020; Garbe et al., 2020; Albrecht et al., 2020; Zeitz et al., 2022; Bochow et al., 2023) and (2) the use of uniform temperature anomalies (e.g., Robinson et al., 2012; Pollard and DeConto, 2012; Cuzzone et al., 2019; Aschwanden et al., 2019; Garbe et al., 2020; Zeitz et al., 2022; Bochow et al., 2023). In this study, we focus on the effects of assumptions made on the forcing fields rather than the validity of the SMB parameterisation methods themselves.

The expected increase in precipitation with increasing surface temperatures is based on the Clausius-Clapeyron relationship (Clausius, 1850), which describes the increase in the saturation water vapour pressure with increasing temperature according

to

$$\frac{\mathrm{d}e_s}{\mathrm{d}T} = \frac{L_T e_s}{RT^2}\,, \tag{1}$$

with the saturation vapour pressure $e_s$, the temperature $T$, the specific latent heat of evaporation of water at $T = 0°\mathrm{C}$ $L_T = 2.5 \cdot 10^6\,\mathrm{Jkg}^{-1}$ (Henderson-Sellers, 1984), and the specific gas constant for water vapour $R = 462\,\mathrm{JK}^{-1}\,\mathrm{kg}^{-1}$. The saturation water vapour pressure thus increases approximately exponentially with temperature. Assuming the precipitation $P$ is solely

governed by the saturation water vapour pressure, then also $P$ is expected to increase exponentially. Using the chain rule, Equation 1 can be rewritten as

$$k = \frac{\mathrm{d}\ln(e_s)}{\mathrm{d}T} = \frac{L_T}{RT^2}, \tag{2}$$

with the growth constant $k$.

Using Equation 2 for a simple estimate of the precipitation sensitivity already gives a deviation from the commonly used

value of 7-8%. For inland Greenland the approximate annual near-surface temperature between 1996 and 2019 is $-23°\mathrm{C}$ (Jiang et al., 2020). Plugging the given values into Equation 2 gives $k = 0.086\,1/\mathrm{K}$, which corresponds to a precipitation sensitivity of approximately 9%/K.

In the literature a precipitation sensitivity between 4 and 8%/K is commonly used for simulating the future evolution of the Greenland ice sheet (Huybrechts, 2002; Robinson et al., 2012; Frieler et al., 2015; Saito et al., 2016; Goelzer et al., 2020; Zeitz

et al., 2021; Aschwanden et al., 2019; Zeitz et al., 2022; Bochow et al., 2023). However, these values are often based on older generations or a limited selection of climate models (Robinson et al., 2012; Frieler et al., 2015; Aschwanden et al., 2019; Zeitz et al., 2022; Zhang et al., 2024), and sensitivities are derived using inaccurate regions for Greenland. For example, they often include Iceland or parts of Canada, which can substantially influence the sensitivity (Frieler et al., 2012). Additionally, the precipitation parameterisation as well as uniform temperature anomalies assume that there is a uniform change of temperatures

and precipitation rates across Greenland, which is unlikely to be true. It has been shown using observations and models that, regionally, the deviations from the thermodynamic expectation around 7%/K can be highly significant (Traxl et al., 2021; Nicola et al., 2023).

Here, we give an update on some commonly used parameterisation factors, informed by the CMIP6 projections (Eyring et al., 2016). We analyse temperature and precipitation changes in Greenland from the year 1850 to 2100 for four future Shared Socio-

economic Pathways scenarios (SSPs) using the CMIP6 ensemble: SSP1-2.6 (Low challenges to mitigation and adaptation), SSP2-4.5 (Medium challenges to mitigation and adaptation), SSP3-7.0 (High challenges to mitigation and adaptation) and SSP5-8.5 (High challenges to mitigation, low challenges to adaptation) (Riahi et al., 2017; Allan et al., 2021). Based on this analysis, we derive uniform and spatially resolved temperature scaling factors and precipitation sensitivities for Greenland. To explain differences in the individual model responses, we investigate the influence of the respective model resolution and

changes in the Atlantic meridional overturning circulation (AMOC) on the near-surface temperature and precipitation changes. Subsequently, we show the difference between using spatially uniform and spatially resolved temperature and precipitation anomalies, as well as the corresponding sensitivities, for the short- and long-term evolution of the GrIS, using the Parallel Ice

Sheet Model (PISM) with the simple diurnal energy balance model (dEBM-simple) (Winkelmann et al., 2011; Zeitz et al., 2021).

## 2 Data and methods

### 2.1 CMIP6

We utilise 32 models from CMIP6, which were all the models for which simulations could could be downloaded during the data collection (see Appendix). In addition to the historical runs, we use four SSP scenarios: SSP1-2.6, SSP2-4.5, SSP3-7.0, and SSP5-8.5 (Riahi et al., 2017; Allan et al., 2021).

We analyse the model-specific and ensemble mean of the historical (1850-2015) and future atmospheric near-surface temperature (tas) and precipitation (pr) changes in Greenland until the year 2100. For the analysis, we regrid all models to a common Gaussian grid with a resolution of $0.5° \times 0.5°$ using a bilinear grid interpolation, using the cdo command line tools (Schulzweida, 2023). For all models, we define Greenland as the geopandas area *Greenland* with a buffer distance of 0.5 toward the outside corresponding to approximately $50\,\mathrm{km}$ (Jordahl et al., 2020). With this approach we avoid the problem of varying land-sea masks that arise due to the different native grids and the regridding process. We verify our geopandas approach by checking the land-ice fraction variable (sftgif) for the models that provide this variable, that is, 19 out of the 33 investigated models. We find an average land-ice cover of 77% with a minimum value of 63% and a maximum value of 85% for our masked region.

We obtain spatiotemporally resolved as well as uniform scaling factors for the near-surface temperatures and precipitation against the global mean temperature (GMT) and time. For the uniform scaling factors, we use a linear fit of the spatially weighted annual and seasonal (winter and summer) near-surface temperatures and precipitation rates in Greenland against the respective model GMTs. For the precipitation we follow a similar approach to Nicola et al. (2023) and fit the log-scaled precipitation against the respective temperatures, motivated by the Clausius–Clapeyron relationship. We define the sensitivity of precipitation for each degree of warming $s$ as

$$s = 100\frac{\%}{\mathrm{K}} \cdot (e^{k/\mathrm{K}} - 1), \tag{3}$$

where $k/\mathrm{K}$ is the unitless growth factor from Equation 1. This follows from the assumption that the precipitation $P$ increases exponentially with the sensitivity $s$ according to

$$P = P_0 \left(1 + \frac{s}{100\%/\mathrm{K}}\right)^{\Delta T/\mathrm{K}} \tag{4}$$

for a temperature change $\Delta T$. Then a linear regression of $\ln(P)$ against $T$ directly gives $k$ as a fit parameter and $s$ can be calculated according to Equation 3. It has to be noted that $k$ is sometimes directly defined as the precipitation sensitivity (Held and Soden, 2006; Nicola et al., 2023). However, for small $k < 0.1$, $k \approx s$ and they can easily be converted between one another.

For the spatial varying scaling factors, we fit the local near-surface temperature and local precipitation values, that is, every grid cell against the GMT. Additionally, we derive scaling factors of the local precipitation against local near-surface temperature (Fig. I1).

## 2.2 PISM Model description

We use the Parallel Ice Sheet Model (PISM) in the version v2.1-1-g6902d5502 (Winkelmann et al., 2011) with the dEBM-simple surface mass balance module (Krebs-Kanzow et al., 2018, 2021; Zeitz et al., 2021). This model configuration has successfully been used for future projections of the GrIS before (Zeitz et al., 2021; Bochow et al., 2023) and has been shown to realistically reproduce past ice sheet states (Zeitz et al., 2021; Garbe et al., 2023). We follow the ice sheet initialisation and setup from Bochow et al. (2023) with some different choices for model parameters that we explicitly mention in the following.

PISM is a thermomechanically coupled ice sheet model using the shallow-shelf (SSA) and shallow-ice approximation (SIA). The ice rheology is based on the Glen–Paterson–Budd–Lliboutry–Duval flow law (Lliboutry and Duval, 1985) with an exponent of $n = 3$ and the enhancement factors $E_{\mathrm{SIA}} = 3$ and $E_{\mathrm{SSA}} = 1$.

Furthermore, we use a pseudo-plastic sliding law (Schoof and Hindmarsh, 2010)

$$\boldsymbol{\tau}_{\mathrm{b}} = -\tau_{\mathrm{c}} \frac{\mathbf{u}}{u_0^q |\mathbf{u}|^{1-q}}, \tag{5}$$

with the basal shear stress $\boldsymbol{\tau}_{\mathrm{b}}$, the basal sliding velocity $\mathbf{u}$, the yield stress $\tau_{\mathrm{c}}$ and a threshold velocity $u_0 = 100\,\mathrm{m/yr}$ and the exponent factor $q = 0.5$. The yield stress $\tau_{\mathrm{c}}$ is given by the Mohr-Coloumb criterion (Cuffey and Paterson, 2010)

$$\tau_{\mathrm{c}} = c_0 + (\tan\phi)\, N_{\mathrm{till}} \tag{6}$$

with the effective pressure $N_{\mathrm{till}}$, the till friction angle $\phi$ and and the till cohesion $c_0 = 0$. The till friction angle $\phi$ is a piece-wise linear function of the bed elevation and the effective pressure $N_{\mathrm{till}}$ is determined by the subglacial hydrology model. We use the Lingle-Clark Earth deformation model with a lithosphere flexural rigidity of $5 \cdot 10^{24}\,\mathrm{Nm}$, mantle density of $3300\,\mathrm{kg/m^3}$ and a mantle viscosity of $1 \cdot 10^{24}\,\mathrm{Pas}$. We apply a spatially uniform lapse rate of $6\,\mathrm{K/km}$ across the whole ice sheet. The precipitation scales with the surface-height-induced near-surface temperature change by a factor of $5\,\%/\mathrm{K}$. We use a horizontal resolution of $10\,\mathrm{km}$ and a vertical resolution of $40\,\mathrm{m}$. Calving is modelled as a combination of prescribed front-retreat calving based on the observed present-day extent of the GrIS and von Mises calving with constant calving parameters (Morlighem et al., 2016). The prescribed front retreat calving means that the ice sheet is not allowed to grow beyond the prescribed present-day extent.

To calculate the surface mass balance we utilise dEBM-simple (Zeitz et al., 2021). The dEBM-simple is a standalone model that requires (monthly) near-surface air temperature and total precipitation fields to calculate the surface energy balance. The total melt is given by the sum of the insolation driven melt $M_I$, the temperature driven melt $M_T$ and the melt offset $M_O$:

$$M_I = \frac{\Delta t_\Phi}{\Delta t \rho_{\mathrm{w}} L_{\mathrm{m}}} \tau_{\mathrm{A}} (1 - \alpha) \bar{S}_\Phi, \tag{7}$$

$$M_T = \frac{\Delta t_\Phi}{\Delta t \rho_{\mathrm{w}} L_{\mathrm{m}}} c_1 T_{\mathrm{eff}}, \tag{8}$$

$$M_O = \frac{\Delta t_\Phi}{\Delta t \rho_{\mathrm{w}} L_{\mathrm{m}}} c_2, \tag{9}$$

with the fraction of the day during which the sun is above the elevation angle ($\Phi$) $\frac{\Delta t_\Phi}{\Delta t}$, the latent heat of fusion $L_m$, the transmissivity of the atmosphere $\tau_A$, the surface albedo $\alpha$, the mean top of the atmosphere insolation during which the sun is above the elevation angle $\bar{S}_\Phi$, the melt parameters $c_1$ and $c_2$, the density of water $\rho_w$ and the effective temperature $T_{eff}$. The average melt is approximated by

$$
M = \begin{cases} M_I + M_T + M_O, & T \geq T_{\min}, \\ 0, & T < T_{\min}. \end{cases} \tag{10}
$$

with the threshold temperature $T_{\min}$ below which no melt occurs. The effective temperature $T_{eff}$ is the expected value of the temperature fluctuation above the positive threshold temperature $T_{pos}$ and is assumed to follow a Gaussian distribution similar to common PDD methods, given by

$$
T_{eff}(T,\sigma) = \frac{1}{\sigma\sqrt{2\pi}} \int\limits_{T_{pos}}^{\infty} \xi \exp\left( -\frac{(\xi - T)^2}{2\sigma^2} \right) d\xi \tag{11}
$$

with the standard deviation $\sigma$. For the dEBM-simple melt equation, we use the parameters $c_1 = -90\,\mathrm{W/m^2 K}$ and $c_2 = 30\,\mathrm{W/m^2}$. The mean top of the atmosphere insolation $\bar{S}_\Phi$ is a function of the orbital parameters and for our simulations, we fix the orbital parameters to present-day values. For a full description of dEBM-simple we refer to (Zeitz et al., 2021) and the PISM documentation (Winkelmann et al., 2011).

## 2.3 Model spinup

We spin up the ice sheet to a close to present-day state by bootstrapping the model, that is heuristically filling in the missing fields from observed present-day conditions (see PISM documentation Winkelmann et al. (2011)), including ice thickness, bedrock elevation (Morlighem et al., 2017) and basal heat flux (Shapiro and Ritzwoller, 2004), and let it reach equilibrium. The temperatures at depth are determined by solving a one-dimensional steady-state differential equation, which balances conduction and vertical advection, and the vertical velocity is used to interpolate linearly from the surface mass balance rate at the top to zero at the bottom (Winkelmann et al., 2011). We run the model for 150,000 years with a fixed climate to reach a close to present-day equilibrium ice sheet state. We force the model with the climatological monthly mean near-surface temperature and precipitation fields from the years 1980-2000 from the regional climate model MARv3.12 forced by ERA-5 boundary conditions (Fettweis et al., 2017). The thickness and the velocity anomaly compared to observational data are depicted in Fig. R1. Our initial state has an ice thickness root-mean-squared error (rmse) of 228 m compared to observational data (Morlighem et al., 2017; Morlighem, 2022) and a rmse of 112 m/year for the velocity.

## 2.4 Experiment setup

We force the ice sheet model PISM with spatially resolved and uniform temperature anomalies and precipitation sensitivities derived from the CMIP6 ensemble to show the influence of spatially uniform anomalies and sensitivities on the short- and

long-term behaviour of the GrIS. However, it is important to note that we rather run a set of idealised experiments and do not aim to give prognostic sea-level rise estimates.

For the short-term model forcing, we use the average near-surface temperature anomalies and precipitation sensitivities for each scenario compared to the time period 1980-2000. First, we calculate the CMIP6 ensemble average fields for each scenario and the historical time period 1980-2000 to have the same reference period as the MARv3.12 forcing fields that we use as background climate. Subsequently, we calculate the monthly near-surface temperature anomalies by subtracting the historical CMIP6 climatology (1980-2000) from the ensemble averages until the year 2100. To obtain the spatially variable precipitation sensitivities, we divide the precipitation fields by the CMIP6 climatologies (1980-2000). Afterwards, we regrid the fields to the 10 km PISM grid using a first-order conservative algorithm (nco) (Zender, 2008). Additionally, we smooth the obtained regridded fields using the cdo smooth command (Schulzweida, 2023) with a smoothing radius of 40 km to avoid artefacts stemming from the interpolation. Thereby, we obtain spatially resolved monthly anomaly fields of the near-surface temperature, and spatially variable sensitivities factors for the precipitation from 2015 to 2100. For the uniform forcing, we calculate the spatially weighted means of the regridded and smoothed monthly near-surface temperature anomaly fields and precipitation sensitivities that we used for the spatial forcing. For simplicity, we use the precipitation sensitivities directly derived from the precipitation anomalies instead of calculating the precipitation sensitivities based on the near-surface temperature anomalies. While we calculate the anomalies and sensitivities with respect to the CMIP6 climatologies, they are applied to the MARv3.12 near-surface temperature and precipitation fields that we use as background climate for the PISM runs. For the long-term model forcing, we calculate the 10 year average of the monthly near-surface temperature anomalies and precipitation sensitivities from 2090-2100 to avoid any seasonal noise or outliers in the forcing.

There are two different precipitation corrections used in PISM. First, the change in precipitations due to changes in the background climate, that is, changes in the near-surface temperatures without considering changes in the ice sheet geometry. Second, changes in the precipitation due to changes in the ice sheet height and therefore changes in the near-surface temperature due to the lapse-rate. PISM does not natively support spatially and temporally variable lapse-rates or height-change induced precipitation sensitivities. Hence, the lapse-rate and height-change induced precipitation sensitivity are usually constant in time and space in PISM. However, we implement rudimentary spatially variable height-change induced precipitation sensitivities. For this, we manually calculate the precipitation and temperature changes that result from changes in the ice sheet height. Specifically, we run PISM for one year, calculate the height change compared to the previous year and apply a spatially and temporally uniform lapse-rate of 6 K/km. This gives the temperature correction due to changes in the ice sheet height, which we apply as near-temperature anomalies in the following year. Subsequently, we calculate the scenario-specific precipitation correction by multiplying the monthly-resolved initial precipitation given by MARv3.12 with the spatially resolved local precipitation sensitivities (Fig I1) given the height-change induced temperature changes. Lastly, we restart PISM with the new updated fields. This method is computationally not very efficient, therefore we only run a limited set of short-term simulations (1,000 years).

In total, we run three different sets of experiments varying between spatially variable and uniform near-surface temperature anomalies, precipitation sensitivities, and height-change induced precipitation sensitivities (Tab. 1). We only show the simu-

lations with a lapse rate of 6 K/km and a height-change induced precipitation sensitivity of 5%/K for the experiments with uniform lapse rates and sensitivities in the main figures. For plots showing the ice volume, we normalise the ice volume such that the initial volume corresponds to the observed ice volume of 7.42 m sea-level equivalent (Morlighem et al., 2017).

## 3 Results

### 3.1 Greenland's climate in CMIP6

#### 3.1.1 Temperature projections in CMIP6

The projected rise in the GMT in CMIP6 is generally higher than in the previous model intercomparison CMIP5 (Tebaldi et al., 2021). For Greenland, most CMIP6 models have a higher near-surface temperature sensitivity than the corresponding CMIP5 models (Zhang et al., 2024). The near-surface temperatures in Greenland are relatively constant in all models until the year 1980 (Fig. 1a). However, there is an accelerated rise in the near-surface temperatures between 1980 and 2100. Only the most optimistic SSP1-2.6 scenario shows a relatively constant near-surface temperature in Greenland after 2050. We find an ensemble mean near-surface temperature rise in Greenland of $\Delta T = 3.27 \pm 1.50°C$ (SSP1-2.6), $4.98 \pm 1.75°C$ (SSP2-4.5), $6.73 \pm 2.06°C$ (SSP3-7.0) and $8.15 \pm 2.30°C$ (SSP5-8.5) above the pre-industrial (1850-1900) level by 2090-2100 (Fig. 1a). There is a considerable spread in the model response, with one model even predicting a near-surface temperature decrease by 2100 under the SSP1-2.6 scenario (Fig. 1a). It is important to note that all near-surface temperature and precipitation projections from the CMIP6 ensemble assume a non-evolving topography of the ice sheet, that is, changes in the ice sheet geometry are not considered in these simulations.

In all scenarios, the ensemble mean of the annual mean near-surface temperature responds in good approximation linearly to the increase in the GMT (Fig. 2a). We find scaling factors ranging from $s = 1.29$ (SSP1-2.6), $s = 1.49$ (SSP2-4.5), $s = 1.54$ (SSP3-7.0) and $s = 1.53$ (SSP5-8.5) of the annual regional near-surface temperature against the GMT until 2100 (Fig. 2a), suggesting a state-dependence of the scaling factor. The scaling factor (or sensitivity) can be interpreted as a direct analogy of Arctic amplification, but for a Greenland specific context. The warming is generally stronger in the winter season (DJF) with more than 40% faster warming than in the summer (JJA) for all future scenarios, except SSP1-2.6 (Fig. 2b,c). In the SSP1-2.6 scenario, the warming in winter and summer shows the same response with $s = 1.16$. However, the winter near-surface temperatures in SSP1-2.6 do not follow a strong linear trend ($R^2 = 0.46$).

While the annual near-surface temperature in the historical period also shows a linear response, it shows stronger warming of Greenland with $s = 2.0$ compared to the future scenarios. In the historical period, the winter near-surface temperatures increase more than 50% faster than the summer near-surface temperatures. This is less than some previously reported values that indicated twice as much warming in winter than in summer (Robinson et al., 2012). In total, the historical runs show the highest scaling factors for summer, winter, and annual near-surface temperatures out of all scenarios.

There is a considerable inter-model spread in the spatially averaged scaling factors (Fig. A1). Specifically, the SSP1-2.6 scenario shows a large inter-model spread, ranging from 0 to 2.25 in summer and from -2 to 4 in winter (Fig. A1a,b). In other

words, in some models the spatially averaged winter near-surface temperature in Greenland decreases two times faster than the GMT increases, while in other models the near-surface temperature increases 4 times faster than the GMT. In general, the spread in the response of the winter near-surface temperatures is larger than in the summer annual near-surface temperatures (Fig. A1b). Furthermore, the more extreme the future scenario, the smaller the range in the scaling factors, besides the annual near-surface temperatures in the SSP2-4.5 scenario. Additionally, we show the relationship between the winter and summer near-surface temperatures.

In all future scenarios, besides SSP1-2.6, the majority of the models shows a faster increase in the winter near-surface temperatures than in the summer near-surface temperatures, i.e. $s > 1$ (Fig. A1d). However, the uncertainty is considerable. The ensemble mean of the spatially-averaged scaling factors agrees approximately with the scaling factors derived from the ensemble mean of the near-surface temperatures, that is quasi-linearly $\overline{s(T)} \approx s(\overline{T})$. This suggests that the response of the ensemble mean near-surface temperatures is an adequate representation of the mean scaling factors. It also has to be noted that the relationship between the GMT and the spatially averaged seasonal near-surface temperatures for some models does not necessarily follow a clear linear relationship. This is especially visible when comparing the scaling factors for each season against the GMT and the scaling factors of the individual seasons against each other (Fig. A1d). However, since the majority of models shows a strong linear relationship, the quasi-linearity of the ensemble mean generally holds.

There is a clear spatial dependency of the near-surface temperature sensitivity in Greenland (Fig. 2d-g). For the historical period, there is a clear warming gradient between the interior and the margin of the GrIS (Fig. C1a). However, this gradient is inverted in summer compared to winter (Fig. C1f). In winter, the margins of the ice sheet warm considerably faster than the interior, with a scaling factor exceeding 300% at the margin and less than 200-250% in the interior of the GrIS. On the other hand, in the summer most of the interior ice sheet warms up to 150% faster than the global average, with a decreasing scaling factor toward the margin. The gradient in summer is not as pronounced as in winter. These sensitivities are in accordance with observational near-surface temperature records in southwestern Greenland, which show a 2.4 times stronger warming in the winter than in the summer for the time period 1850-2019 (Cappelen, 2020; Bochow et al., 2023). In both seasons, almost the entire GrIS shows stronger T/GMT scaling in the historical period than in the future scenarios.

In the future scenarios, there is a north-south gradient in the ensemble mean annual near-surface temperature with decreasing temperature sensitivity towards the South, rather than a gradient between the interior and the margin of the GrIS as in the historical runs. For SSP1-2.6, all of Greenland approximately south of the Ilulissatfjord shows a sensitivity below 100%, that is, it warms slower than the GMT (black star in Fig. 2d). This 100%-contour migrates southward the more extreme the scenario (Fig. 2d-g). For the SSP3-7.0 and the SSP5-8.5 scenario, only the southernmost part of Greenland shows a slower mean annual warming than the global mean (Fig. 2g) However, there is a pronounced seasonal difference in the spatial response. While the summer near-surface temperatures increase faster in the interior of the GrIS than at the margins, the winter near-surface temperatures show a north-south gradient with considerably more warming in the northern parts of the ice sheet (Fig. C1). While this spatial dependence varies in intensity between the scenarios, it is consistent across all the SSPs.

The inter-model spread in the spatial response, relative to the GMT increase, is large. Here, we only show the spatial differences for the annual near-surface temperatures, but similar patterns are observed for the individual seasons. Especially for the

most optimistic SSP1-2.6 scenario, there is a pronounced difference in the different model responses (Fig. E1). Several models show a near-surface temperature decrease for the 21st century, mostly in the southern and southwestern parts of Greenland. The two models CESM2 and CESM2-WACCM even project a near-surface temperature decrease for most of Greenland except for the northernmost part. This is mostly due to a strong decrease in the winter near-surface temperatures in these models. For the SSP2-4.5 scenario, a minority of models show a decrease in the near-surface temperatures at the southwestern margins (Fig. F1). In the SSP3-7.0 scenario, only one model (FGOALS-g3) shows a decrease in the annual near-surface temperatures at the southern margin (Fig. G1). Interestingly, FGOALS-g3 is the lowest resolution model analysed. Similarly, only the FGOALS-g3 model shows a minimal near-surface temperature decrease at the southernmost margin in the SSP5-8.5 scenario (Fig. H1). In general, the more extreme the scenario, the more uniform the model responses become.

For the historical runs, all models show a clear increase in the mean annual near-surface temperatures in all of Greenland (Fig. D1). However, similar to the future scenarios, there are clear differences in the magnitudes of the temperature change (relative to the model GMT). The EC-Earth3-Veg-LR model shows the strongest response out of all models, with a temperature sensitivity of more than 300% in most of Greenland. In contrast, several models such as NorESM2-LM show a slower warming than the GMT, i.e. a scaling factor smaller than 100%, in some parts of Greenland.

The regional ensemble mean near-surface temperatures show a strong linear dependence on the GMT in all scenarios and seasons, with a mean $R^2 > 0.7$ (Fig. C1,2&B1). Only the winter near-surface temperatures in the SSP1-2.6 scenario (Fig. C1b) show a small coefficient of determination $R^2 = 0.36$. The $R^2$ values show the lowest values in southwestern Greenland (Fig. B1a) with a steady increase towards the north. However, this spatial dependence is only pronounced for the low emission scenarios. The linear relationship between the regional near-surface temperatures and the GMT is also clearly visible in the single model regional scaling factors, especially for the SSP3-7.0 and SSP5-8.5 scenarios, where $R^2 > 0.7$ for most models (Fig. D1-H1).

Some of the analysed models are available in lower and higher resolution versions, such as NorESM2, MPI-ESM1, MIROC-ES2, or EC-Earth3-Veg. This allows for a direct comparison between the same model but different resolutions. While there are some visible differences in the spatial patterns between higher and lower resolution models, these differences cannot be easily attributed to the different resolutions alone. For example, in the SSP5-8.5 scenario, the high resolution version MIROC-ES2H seems to resolve the topography of the GrIS, and hence the temperature response, in the interior of the ice sheet better than the low resolution version MIROC-ESL2 (Fig. H1). However, the lower and higher resolution versions of the same model do not show a systematic difference in the temperature response (Fig. Q1a).

We analyse the relationship between the near-surface temperature changes and changes in the AMOC strength for all the models where information on the AMOC strength is available (between 19 and 21 models) (Baker et al., 2023). The annual near-surface temperature anomaly by 2090-2100 has a moderate correlation with the response of the AMOC ($r = 0.59$ SSP1-2.6; $r = 0.62$ SSP2-4.5; $r = 0.64$ SSP3-7.0; $r = 0.47$ SSP5-8.5) (Fig. 5a). Generally, for each scenario, we find that the stronger the decline of the AMOC in the respective model, the smaller the near-surface temperature increase by the end of 2100 in Greenland. This is not surprising, given that a weakening of the AMOC is expected to have a cooling effect in parts of the Northern hemisphere (Bellomo et al., 2021). The differences in the AMOC decline seem to explain why some models in

moderate scenarios show a stronger near-surface temperature increase in Greenland than other models in the most extreme scenario (i.e. NorESM2-LM in SSP5-8.5 against CanESM5 in SSP2-4.5, Fig. 5a). However, it is difficult to quantify the effect of the AMOC change on the spatial temperature response. For example, the models NorESM2-MM and GFDL-ESM4, which have the same model resolution, show the same average near-surface temperature increase by the year 2100 but NorESM2-MM shows a 50% stronger AMOC decline (Fig. 5). For both models, the spatial near-surface temperature scaling factors are very similar for all seasons (Fig. H1). This implies that the strength of the AMOC decline alone cannot explain the differences in the near-surface temperature change in Greenland.

### 3.1.2 Precipitation projections in CMIP6

Similar to the near-surface temperature projections, the precipitation rates in Greenland are expected to increase in the future, with a stronger response in CMIP6 than in CMIP5 (Zhang et al., 2024). The precipitation anomalies stay constant until 1980, analogous to the near-surface temperature anomalies. In all scenarios, we find an increase in the ensemble mean annual precipitation rates compared to the reference period 1850-1900 (Fig. 1b). However, some models in the SSP1-2.6 and SSP2-4.5 scenarios predict a decrease in the annual precipitation rates at the end of the 21st century. We find an ensemble mean of the annual precipitation increase of $\Delta P = (804 \pm 633)$ mm/year (SSP1-2.6), $1302 \pm 715$ mm/year (SSP2-4.5), $1866 \pm 954$ mm/year (SSP3-7.0) and $2326 \pm 1093$ mm/year (SSP5-8.5) by the year 2090-2100 (Fig. 1b). While the ensemble mean precipitation anomalies show a similar response as the near-surface temperature anomalies, the uncertainty, or ensemble spread, is larger than for the near-surface temperatures.

The log of the ensemble mean annual precipitation shows an increase in all scenarios, including the historical period (Fig. 3a). The highest precipitation sensitivity is observed in the historical period with $s = 8.35\%$/K, followed by the SSP5-8.5 scenario with $s = 6.09\%$/K, SSP3-7.0 with $s = 5.53\%$/K, SSP2-4.5 with $s = 3.74\%$/K and SSP1-2.6 with $s = 3.02\%$/K. However, the relationship between the logarithm of precipitation and GMT is not clearly linear, i.e. the precipitation rates do not necessarily increase exponentially with the GMT. Specifically, for the SSP1-2.6 scenario, the coefficient of determination is only $R^2 = 0.27$, and for the historical period $R^2 = 0.58$. This might be a consequence of the small increase in the GMT in these two scenarios, leading to a larger contribution of the interannual precipitation variability to the observed precipitation changes. For the more extreme scenarios (SSP3-7.0 and SSP5-8.5), the linear relationship between log precipitation and GMT is very clear ($R^2 > 0.9$).

A similar pattern arises for the ensemble means of the summer and winter precipitation changes (Fig. 3b,c). The sensitivity of the summer precipitation is higher in all future scenarios and more uniform across all scenarios compared to the winter sensitivity. We find a summer precipitation sensitivity, relative to the GMT, of $s = 5.75\%$/K (Historical), $s = 6.45\%$/K (SSP1-2.6), $s = 4.6\%$/K (SSP2-4.5), $s = 6.51\%$/K (SSP3-7.0) and $s = 7.03\%$/K (SSP5-8.5) (Fig. 3b). We find a winter precipitation sensitivity of $11.30\%$/K (Historical), $-2.79\%$/K (SSP1-2.6), $2.82\%$/K (SSP2-4.5), $4.37\%$/K (SSP3-7.0) and $4.61\%$/K (SSP5-8.5). This implies a reduction of winter precipitation rates with increasing GMT for the SSP1-2.6 scenario. However, similarly to the changes in the annual precipitation anomalies, the historical, SSP1-2.6 and partially the SSP2-4.5 precipitation sensitivities show a small coefficient of determination for seasonal precipitation (Fig. 3b,c). Additionally, the inter-model spread is

considerable, especially for the winter precipitation sensitivity (Fig. K1c). For example, for the SSP1-2.6 scenario, the standard deviation of the winter precipitation sensitivity is considerably larger than the sensitivity itself. The SSP1-2.6 winter precipitation sensitivity ranges from -29%/K for the NorESM2-LM model to 26%/K for the BCC-CSM2-MR model. It also has to be noted that the $R^2$-coefficient of the spatially averaged precipitation sensitivity is very small for most fits, i.e. $R^2 < 0.1$, especially for the moderate scenarios. This indicates that a spatially and seasonally uniform precipitation sensitivity for the whole GrIS fails to capture spatially heterogeneous patterns.

This is further supported by the spatial patterns of the ensemble mean of the annual precipitation sensitivities (Fig. 3d-g). In all scenarios, there is a clear northeast-southwest gradient in the sensitivity. The northeastern part of Greenland shows the highest annual precipitation increase with increasing GMT in all scenarios. The spatial sensitivity differences are largest in the SSP1-2.6 scenario with $s > 15\,\%/$K in the northeast of Greenland to $s < 0\,\%/$K at the southern and southeastern margin (Fig. 3d). In the SSP2-4.5 scenario, only the southernmost tip of Greenland shows a negative precipitation sensitivity (Fig. 3e). In all other scenarios, the annual precipitation sensitivity is positive, i.e. there is an increase in the precipitation with an increase in the GMT in all of Greenland. The seasonal difference in the spatial responses is pronounced (Fig. J1). In winter (Fig. J1a-e), the precipitation increase is most pronounced in northern Greenland, with less precipitation increase in western and southeastern Greenland. For the SSP1-2.6 scenario, large parts of southern Greenland show a negative precipitation sensitivity. In summer (Fig. J1f-j), eastern Greenland shows the highest sensitivity. In the historical period, the spatial sensitivity patterns are relatively similar to the other scenarios (Fig. J1a,f). However, western and northeastern Greenland show a slightly lower precipitation sensitivity in the historical period for both winter and summer than the future scenarios (Fig. J1). At the same time, the winter precipitation sensitivity of southeastern Greenland is substantially higher for the historical period. While we show sensitivities of the ensemble mean precipitation rates, the ensemble mean of the single-model sensitivities shows similar patterns and intensities.

Interestingly, the $R^2$-values of the fit of the ensemble mean precipitation rates against GMT are low for the moderate emission scenarios (Fig. B1e,f & J1). At the margins and especially at the southeastern margin, the $R^2$-values are small due to high precipitation variability in these regions (Fig. B1e-h). Similarly, the $R^2$-values of the single model precipitation sensitivities are very low for most models in all scenarios, i.e. $R^2 < 0.2$, indicating that there is not a clear linear relationship between $\ln(P)$ and the GMT or mean near-surface temperatures in Greenland in individual models (Fig. L1-P1). However, for the SSP3-7.0 and SSP5-8.5 scenario, the logarithm of the ensemble mean annual precipitation rates shows a clear linear relationship with the GMT ($R^2 > 0.77$) (Fig. B1g,h).

Since the spatially averaged annual and seasonal near-surface temperatures in Greenland show a strong linear relationship with the GMT (Fig. 2a-c), the precipitation sensitivities relative to the GMT can easily be converted to precipitation sensitivities relative to the spatially averaged regional Greenland near-surface temperatures. The resulting sensitivities are slightly lower due to the accelerated increase in near-surface temperatures in Greenland compared to the GMT, but do not show a substantial change in the spatial patterns (Fig. I1).

The inter-model spread in the spatial response is large for the summer precipitation and for the winter precipitation in all scenarios. Several models show strong variation from the cross-model mean of the summer precipitation sensitivity, with

predicted precipitation decreases in large parts of Greenland, especially for the moderate scenarios. This is also visible in the annual precipitation sensitivities (Fig. L1-P1). The spatial response differs widely between the single models, especially for the SSP1-2.6 scenario (Fig. M1). Generally, there are some differences in the spatial patterns between the low- and high-resolution versions of the same model and the high-resolution versions seem to resolve local effects better (Fig. Q1b). For example, in the SSP5-8.5 scenario, MPI-ESM1-2-HR predicts a lower, even negative, precipitation sensitivity at the southeastern margin of the GrIS, while the LR version does not (P1). Similar differences are visible for NorESM2-LM and NorESM2-MM and the MIROC-ES2H and MIROC-ES2L versions in the SSP5-8.5 scenario. However, for instance, CNRM-CM6-1 shows the opposite behaviour. Additionally, the lowest resolution model analysed, FGOALS-g3, shows a very similar precipitation sensitivity pattern in the SSP5-8.5 scenarios as some of the highest resolution models such as MPI-ESM1-2-HR.

Similar to the near-surface temperature, we find a relationship between the precipitation anomalies and the change in AMOC strength by the year 2100 ($r > 0.62$ for all scenarios, Fig. 5b). The stronger the AMOC decline, the less the precipitation rates increase in Greenland. Analogously, the precipitation sensitivity is smaller the stronger the decline in the AMOC strength, in the respective scenario (Fig. 5c). In other words, the precipitation increases less for each degree of warming if the AMOC shows a stronger weakening. This is in accordance with previous studies, that show that a strong AMOC decline has a negative effect on precipitation in Greenland, especially in the wetter regions in southern Greenland and particularly at the southeastern margin (Bellomo et al., 2021). Consequently, this effect is also visible in the spatial precipitation sensitivities. For example, 4 out of 5 available models with an AMOC decline $> 13\,\mathrm{Sv}$ in the SSP5-8.5 scenario show a clear negative precipitation sensitivity at the southeastern margin (CESM2-WACCM, FGOALS-g3, MRI-ESM2-0 and NorESM2-MM, Fig. P1&5c). However, while NorESM2-LM shows the strongest AMOC decline in the SSP5-8.5 scenario among all models with available AMOC information, the precipitation sensitivity is negative only in a small part of the southeastern margin (Fig.P1). Similarly, ACCESS-CM2 shows a slightly stronger AMOC decline in the SSP5-8.5 scenario than CNRM-ESM2-1, but does not show a negative precipitation sensitivity in southern Greenland, in contrast to CNRM-ESM2-1.

## 3.2 Modelling the response of the ice sheet

To show the influence of the modelling choices, we run simulations with the ice sheet model PISM-dEBM-simple with spatially resolved and spatially uniform scaling factors and sensitivities for all analysed SSP scenarios.

### 3.2.1 Short-term response

We initialise the ice sheet to a close to present-day state corresponding to the year 2015. Subsequently, we run the model with the dEBM-simple surface mass balance module (Zeitz et al., 2021), which only needs precipitation and near-surface temperatures as input until the year 2100. We keep the orbital parameters fixed in all runs to exclusively extract the influence of the sensitivity choices (c.f. 2.2). In total, we run three different experiments:

(i) uniform scaling factors and sensitivities with constant height-change-induced precipitation sensitivities (Uniform Anomaly/Sens.),

(ii)  spatially resolved scaling factors and sensitivities and with constant height-change-induced precipitation sensitivities and (Spatial Anomaly/Sens.)

(iii)  spatially resolved scaling factors and sensitivities and with spatially varying height-change-induced precipitation sensitivities (Spatial Anomaly/Sens. + sp. height-induced).

For details on the experimental setup, we refer to Section 2.2.

There is a clear difference between the ice sheet volumes after 85 years between experiment (i) and experiments (ii/iii) (Fig. 4a). The runs begin to diverge after approximately 15 to 20 years. In each case, the simulations with a spatially uniform scaling factor (dashed lines in Fig. 4a) show a smaller ice volume than the corresponding simulations with spatially resolved scaling factors (dash-dotted and solid lines). The difference is more pronounced the more extreme the scenario. In the SSP1-2.6 scenario, the ice volume difference between the different experiments is less than 0.5 cm global sea-level equivalent in the year 425   2100, while in the SSP5-8.5 scenario the difference corresponds to ca. 1 cm global sea-level equivalent (Fig. 4a). Interestingly, the difference in total ice volume after 85 years between experiment (ii) and (iii) is almost unnoticeable.

The absolute height-change by the end of the simulation is greatest at the margin, exceeding 100 m in large parts of the margin in the most extreme scenario (Fig. S1). The spatial differences between the experiments show the same pattern for each scenario (Fig. 4b-e). The ice sheet thickness is generally smaller at the southwestern margin when the model is forced 430   with the uniform scaling factors and sensitivities (blue in Fig. 4b-e). In contrast, the ice thickness at the southeastern margin, and partially in the interior of Greenland, is larger for the simulations with uniform scaling factors and sensitivities for each scenario (red). The spatial differences between experiments (ii) and (iii) are in the magnitude of less than 10 m in most parts of the GrIS and are not shown here. The height differences between experiment (ii) and (iii) are more pronounced in some peripheral parts of the ice sheet. These areas are those where, on a decadal time scale, the influence of different height-change-435   induced precipitation sensitivities is expected to be greatest. The height differences between the spatially varying and uniform setups are more pronounced the more extreme the scenario. For the moderate scenarios SSP1-2.6 and SSP2-4.5, the difference in the ice thickness after 85 years is in the magnitude of 10 m to 20 m (Fig. 4b,c). For the SSP5-8.5 scenario, the ice thickness difference is more than 30 m at the southeastern and southwestern margins, and partially extends further into the ice sheet.

This observed influence of the scaling factors is in accordance with the spatial sensitivities shown before. The southwestern 440   margin of the GrIS shows a smaller near-surface temperature increase than the average GrIS, especially for the annual melt period (summer) (Fig. C1f-j). This leads to an overestimated melt rate in this region for the uniform temperature anomalies. In contrast, the precipitation sensitivity at the southwestern margin is close to the GrIS-wide average (Fig. J1). Similarly, the temperature-driven melt is larger for the spatially resolved anomalies at the northern ice sheet margins. For the southeastern margin, the changes in the simulated ice sheet thickness are mostly due to the differences in the precipitation sensitivities. The 445   precipitation increase per degree of warming is smaller at the southeastern margin than the GrIS-wide average (Fig. 3d-g). In the SSP1-2.6 scenario, there is even a decrease in the precipitation rates. This leads to less accumulation and hence reduced SMB in this region compared to the uniform sensitivities. In fact, the ice thickness differences agree very well with the spatial patterns of the precipitation sensitivity in eastern Greenland observed by the year 2100.

To investigate the dynamic contribution to the ice thickness changes, we follow the approach by Goelzer et al. (2020). We calculate the time-integrated SMB anomaly for the whole time span and calculate the difference from the total local mass change (residual height change, Fig. S1). To calculate the SMB anomaly, we run a control run without any temperature or precipitation anomalies or height-induced effects and take the difference between the SMB from the control run and the respective scenario run. This gives all ice thickness changes that are not directly related to changes in the SMB. We find a positive dynamic contribution to the ice thickness along large parts of the ice sheet margin. This can be explained by dynamic thickening in response to the negative SMB anomalies that steepen the surface slopes at the margin (Huybrechts and Wolde, 1999; Goelzer et al., 2020). Further inwards, the corresponding thinning is visible. The dynamic contribution is generally slightly greater in the simulations with uniform sensitivities and anomalies than for the experiments with spatially resolved sensitivities and anomalies.

### 3.2.2 Long-term response

We extend the short-term simulations for another 100,000 years with temporally constant monthly near-surface temperature anomalies and precipitation sensitivities, derived from the average 2090-2100 CMIP6 climatologies. Due to computational constrains, we extend the runs with spatially varying height-induced precipitation sensitivities (experiments (iii)) only for another 1,000 years.

The long-term response of the ice sheet varies substantially between spatially resolved and uniform scaling factors (Fig. 6 & 7). Similar to the short-term response, the ice volume is generally smaller for uniform near-surface temperature anomalies and precipitation sensitivities than for spatially resolved anomalies and sensitivities (Fig. 6). The long-term differences are most pronounced for the SSP2-4.5 scenario. In this scenario, the ice sheet loses almost 60% of its initial volume with uniform anomalies and sensitivities, followed by a temporary recovery and subsequent oscillations in the ice volume. In contrast, for the spatially resolved anomalies, the ice sheet does not lose more than 30% of its initial volume.

For the SSP1-2.6 scenario, the ice sheet response between spatially resolved and uniform anomalies is very similar. The ice sheet loss is limited to 10-25% of the initial ice sheet volume in both cases. While the simulations with uniform anomalies generally show a 5-10% smaller ice volume than the corresponding simulations with spatial anomalies, both simulations show oscillations of the ice sheet on decamillennial time scales that almost seem synchronised. These quasi-periodic decamillennial oscillations of the ice sheet have been observed before and are believed to be a nonlinear interplay between the glacial isostatic adjustment (GIA) and the melt elevation feedback (Zeitz et al., 2022; Bochow et al., 2023; Petrini et al., 2023). The reduced ice sheet volume due to increased near-surface temperatures leads to less load on the bedrock below the ice sheet. Subsequently, the bedrock lifts up and leads to decreasing near-surface temperatures at the ice surface, enabling partial regrowth of the ice sheet, which in turn depresses the bedrock again, closing the feedback loop.

For the SSP3-7.0 and SSP5-8.5 scenarios, the ice sheet is lost almost completely without any recovery, independently of choosing spatially resolved or uniform scaling factors. However, the time scale of the loss differs between the uniform and spatially resolved anomalies. For the SSP3-7.0 scenario, the ice sheet is completely lost (ice volume below $0.25 \cdot 10^{15} \, \mathrm{m}^3$) after 12,400 years (spatially resolved anomalies) and after 7,800 years (uniform anomalies), respectively. Similarly, the ice sheet is

lost after 6,300 years (spatially resolved anomalies) and after 4,000 years (uniform anomalies), respectively, in the SSP5-8.5 scenario. The differences in ice volume after 1,000 years between experiments (ii) and (iii), that is, with or without spatially resolved height-change-induced precipitation sensitivities, are negligible compared to the differences between (i) and (ii/iii). For the high emission scenarios, the runs diverge slightly after circa 750 years, with (iii) generally showing less ice loss, while the differences are almost indiscernible in the low emission scenarios.

The spatial evolution of the ice sheet shows similar patterns for spatially resolved and uniform scaling factors (Fig. 7). However, there are differences in the time scale and extent of the ice sheet retreat. For the spatially resolved anomalies, the southwestern part of the ice sheet is most sensitive to warming, followed by a gradual retreat from the south to the north of the GrIS with a rather abrupt loss of the remaining northern part. In the SSP2-4.5 scenario with spatially varying anomalies, the southeastern part regrows due to the GIA, simultaneously with the loss of the remaining northern part of the GrIS, which is also visible in the minimum extent during the 100,000 years of simulation (red line in Fig. 7b). This leads to the observed quasi-oscillatory behaviour in the ice volume (Fig. 6). Similarly, the southwestern GrIS is most sensitive to warming in the case of uniform anomalies. However, in contrast to spatially resolved anomalies, the northern part retreats almost simultaneously with the southern part of the GrIS.

For the SSP1-2.6, SSP3-7.0 and SSP5-8.5 scenarios, the spatial extent of the ice sheet after 100,000 years is similar for uniform and spatially resolved anomalies (Fig. 7). Furthermore, the minimum extent during the whole simulation is very similar between spatially resolved and uniform scaling factors. Only the SSP2-4.5 scenario shows a very different minimum extent and extent at the end of the simulation (Fig. 7b,f), which is also visible in the ice volume (Fig. 6). The interplay between GIA and melt-elevation feedback becomes more important close to the critical temperature threshold, beyond which the ice sheet melts completely. While the near-surface temperatures are high enough to melt large parts of the ice sheet, the subsequent bedrock rebound still allows partial regrowth of the ice sheet. The SSP2-4.5 anomalies are close to this threshold, therefore the difference between the uniform and spatially resolved near-surface temperature and precipitation anomalies is especially pronounced. In contrast, for the SSP3-7.0 and SSP5-8.5 scenario, the GIA does not allow a regrowth anymore.

To show the influence of different lapse-rate values and height-change induced precipitation sensitivities, we vary both parameters and run ensemble simulations for 100 kyr (Fig. T1). Generally, the higher the lapse-rate the faster and the greater the ice loss (Fig. T1c,d). Unsurprisingly, the lower the height-change induced precipitation sensitivity, the slower and smaller the ice loss. Generally, the qualitative behaviour is the same for different parameter combinations besides the magnitude of ice loss. Only for a lapse rate of 0 K/km, the ice sheet stabilises after some time even for the high-emission scenarios (Fig. T1a). The simulations with uniform anomalies and sensitivities show a faster and greater ice loss than the corresponding simulations with spatially varying anomalies and sensitivities.

## 4   Conclusions & Discussion

We analysed the precipitation and near-surface temperature changes in Greenland for the time period from the year 1850 until the year 2100, based on the output of the latest coupled model intercomparison project (CMIP6). We find a temperature

sensitivity of $s = 1.29$ (SSP1-2.6), $s = 1.49$ (SSP2-4.5), $s = 1.54$ (SSP3-7.0) and $s = 1.53$ (SSP5-8.5) between annual mean near-surface temperatures in Greenland and the GMT. Additionally, we find a sensitivity of the mean annual precipitation rates to GMT between $s = 3.02\,\%$/K (SSP1-2.6), $s = 3.74\,\%$/K (SSP2-4.5), $s = 5.53\,\%$/K (SSP3-7.0) and $s = 6.09\,\%$/K (SSP5-8.5). We find a clear seasonal dependency, both for the temperature scaling and the precipitation sensitivities. Our precipitation
sensitivities, with respect to local near-surface temperatures, are below the theoretically derived value from the Clausius-Clapeyron relationship of ca. 9%/K, in accordance with other studies (Robinson et al., 2012; Frieler et al., 2012, 2015; Nicola et al., 2023; Bochow et al., 2023) (Fig. 3& I1). Our temperature scaling factors are in accordance with an earlier study, which only investigated temperature scaling factors for the historical and SSP5-8.5 scenarios (Bochow et al., 2023). At the same time, our summer temperature scaling factors are higher for all scenarios than a previous estimate based on CMIP3, which estimated
a scaling factor of $s = 0.9 \pm 0.2$ between summer near-surface temperature in Greenland and the GMT (Robinson et al., 2012). Similarly, our winter temperature scaling factors are lower than reported by Robinson et al. (2012). Robinson et al. (2012) report a 2.1 times faster warming in winter compared to the summer, while we only find a ca. 1.4 times faster warming during the winter than during the summer. These differences are very likely due to the substantial changes in the climate models between CMIP3 and CMIP6.
Interestingly, the average regional warming in Greenland by the year 2100, as predicted by the CMIP6 models, is close or even above the critical temperature threshold of the GrIS, which has been estimated to lie between $0.8°$C and $3.0°$C GMT relative to pre-industrial levels (Armstrong McKay et al., 2022). Even for the SSP1-2.6 scenario, the ensemble mean regional warming in Greenland for 2090-2100 is $\Delta T = 3.27°$C above pre-industrial levels (Fig. 1a), which translates to a global warming level of $\Delta T_{\mathrm{GMT}} = 2.5°$C using the corresponding scaling factor for the SSP1-2.6 scenario of $s = 1.29$ (Fig. 2a). Hence,
the critical temperature threshold for the GrIS might be crossed, at least temporarily, even in the optimistic emission scenarios. However, it has been shown that a temporary overshoot of the critical threshold for the GrIS does not necessarily imply a large-scale loss of the ice sheet (Bochow et al., 2023; Höning et al., 2024).

The differences in the precipitation sensitivities between the different emission scenarios and the historical period can be interpreted as a state-dependency of the precipitation sensitivities, that is, the sensitivity is dependent on the background climate
and not only on the regional near-surface temperatures. We attribute this to other global warming-induced changes such as changes in dynamic processes, i.e. atmospheric circulation patterns, that strongly influence precipitation patterns and rates in Greenland (Mernild et al., 2015; Auger et al., 2017; Lewis et al., 2017, 2019; Bellomo et al., 2021). While our approach takes all changes in the precipitation rates into account, we do not distinguish between thermodynamic and dynamically induced changes in the precipitation rates. In other words, our approach assumes that all changes in the precipitation rates are a direct
consequence of temperature change. The low $R^2$ values for individual model fits support the conclusion that the changes in precipitation are not necessarily driven by temperature changes alone, but rather by warming-induced changes in, for example, atmospheric circulation patterns. Especially for the moderate emission scenarios, the low $R^2$ values are a consequence of strong precipitation variability for low levels of warming. For the high emission scenarios, this interannual variability is masked by the strong warming-induced changes. It has been shown that the North Atlantic Oscillation, Atlantic Multidecadal Oscillation
and the Icelandic low have significant correlation with observational precipitation rates across Greenland (Mernild et al.,

2015; Auger et al., 2017). Specifically, the eastern margin of Greenland shows a high precipitation variability in CMIP6 and observations, which has been linked to the North Atlantic storm track (Groves and Francis, 2002; Bogerd et al., 2020). Auger et al. (2017) conclude that increased blocking events in a warmer climate would lead to an increase in variability and likelihood of precipitation in southwest Greenland, and to the opposite effect in southeast Greenland. While this does not directly translate to precipitation sensitivities, we find a generally lower or even negative precipitation sensitivity in southeastern compared to southwestern Greenland (Fig. 3d-g). However, the representation of such atmospheric changes in the current generation of climate models is not clear (Delhasse et al., 2021). The observed state dependency of the sensitivities also implies that historical changes might not be an appropriate predictor for future changes. For example, the average regional near-surface temperatures in Greenland show a higher scaling with the GMT in the historical period than in all future scenarios, even though the linear relationship holds across all scenarios. Scaling factors or sensitivities solely derived from historical data might over- or underestimate future responses. Nevertheless, ideally, observational data should be included for parameterisations to complement modelling results.

We find large differences in temperature and precipitation responses across the different CMIP6 models. Possible reasons include differences in model physics and parameterisation schemes and subsequent changes in other large-scale phenomena, such as the AMOC, which influence the climate in the GrIS region. Indeed, we find a relationship between the change in the AMOC strength and the temperature anomalies and precipitation sensitivities. Generally, a stronger AMOC weakening leads to less warming and less precipitation increase in Greenland. Models that predict a strong AMOC decline by the end of the 21st century seem to predict a decline in precipitation rates in the wettest region of the GrIS, i.e., its southeastern margin. This is in accordance with previous studies that show a negative effect of a strong AMOC decline on the precipitation rates in the subpolar North Atlantic region (Liu et al., 2020; Bellomo et al., 2021). The reduction of precipitation in this region for a strong AMOC decline has been associated with a decrease in the evaporation rates over the ocean into the atmosphere and a reduction in the eddy moisture transport (Liu et al., 2020; Bellomo et al., 2021). Bellomo et al. (2021) even suggests that precipitation changes in models with strong AMOC decline are primarily driven by dynamic changes rather than thermodynamic effects. This is in accordance with our findings, which show that the pure thermodynamic effects cannot explain the changes in precipitation alone.

An obvious explanation for some of the observed differences is the model resolution, as higher resolution models should at least resolve the topography of the GrIS better and capture local effects more accurately. While we find some differences in the spatial patterns of the sensitivities between higher and lower resolution versions of the same models, we do not find any systematic differences. That is, we do not find a clear relationship between model resolution and the near-surface temperature and precipitation sensitivity. While it generally seems that the spread in the spatially averaged model response decreases with increasing resolution, both for the near-surface temperature and precipitation, this is hard to verify due to the limited set of models with low resolution. As such, the different model resolutions are not sufficient to explain the differences in the sensitivities across the CMIP6 ensemble.

Given the strong differences in model responses, the use of a limited selection of climate models or even of individual models to estimate precipitation or temperature changes can lead to vastly different results. Similarly, using single climate model runs

as forcing for standalone ice sheet modelling can produce very different outcomes. We therefore argue that the use of single model runs as forcing for ice sheet modelling, especially on long time scales, needs to be treated with caution. We either suggest the use of (i) ensemble mean climate fields as forcing, which minimise the influence of short-term climate variability on the ice sheet, or (ii) ensemble simulations with single model forcing fields. The latter option allows, in principle, a statistical treatment of the different outcomes. However, running ensemble simulations with different forcing is computationally substantially more expensive.

We show that uniform near-surface temperature and precipitation anomalies, that are often used for long-term modelling of ice sheets, can lead to over- or underestimation of the accumulation and ablation rates and hence the surface mass balance. For the GrIS, simulations with uniform near-surface temperature anomalies based on a Greenland-wide average might overestimate melt rates in large parts of the GrIS. On the other hand, spatially and seasonally uniform precipitation sensitivities might lead to an overestimation of accumulation rates in southern Greenland and an underestimation in northern Greenland. The difference between spatially varying and uniform scaling factors is already clearly visible in short-term (century-scale) simulations, but has an even bigger influence on long-term (millennium-scale) simulations. Interestingly, our simulations with spatially resolved height-change-induced precipitation sensitivities (iii) barely show any difference from our simulations with spatially uniform height-change-induced precipitation sensitivities (ii) on the time scales we investigated (1,000 years). This implies that, on a sub-millennial time scale, a uniform value for the height-change-induced precipitation sensitivity is a reasonable modelling assumption. Ultimately, spatially uniform sensitivities and anomalies in simulations might lead to an underestimation of the long-term stability of parts or even the whole GrIS. Similar concerns have been raised about uniform lapse rates, which are often used in ice sheet modelling (Crow et al., 2024). It is known that the lapse rate in Greenland has a seasonal and spatial dependency (Erokhina et al., 2017) and it has been shown that using spatially and temporally varying lapse rates has a substantial influence on the ice sheet evolution (Crow et al., 2024).

While spatially uniform anomalies capture the overall evolution of the GrIS, we have shown that they tend to over- or underestimate the surface mass balance of parts of the ice sheet. They therefore lead to over- or underestimated projections of sea level or long-term stability of the GrIS. As long as fully coupled ESM simulations on long time scales remain computationally challenging, we recommend the use of spatially and monthly resolved anomalies and sensitivities for long-term ice sheet modelling, instead of spatially uniform anomalies.

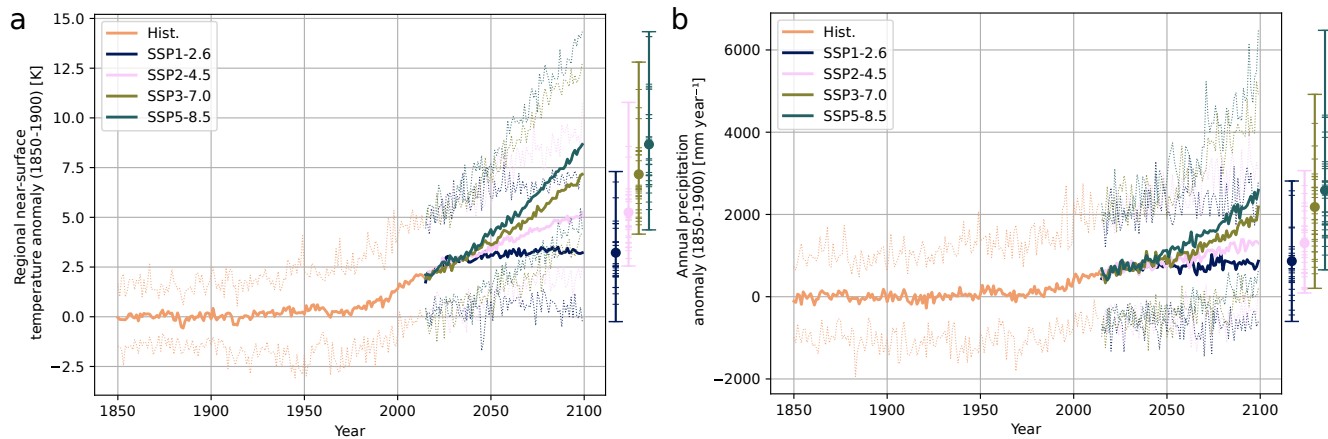

**Figure 1. Ensemble mean of Greenland mean annual near-surface temperature anomalies and precipitation sensitivities in CMIP6 relative to pre-industrial levels.** (a) Ensemble mean of annual near-surface temperature anomalies as predicted by CMIP6 models for all SSP scenarios relative to pre-industrial level (1850-1900). The solid lines denote the ensemble mean, while the dashed lines denote the minimum and maximum anomalies for the respective scenario. The horizontal lines on the bar denote the end-value in the year 2100 for each model, while the dot is the mean of the near-surface temperature in the year 2100. (b) Same as **a** but for the annual precipitation. In each scenario, the ensemble mean of the near-surface temperature and precipitation increase.

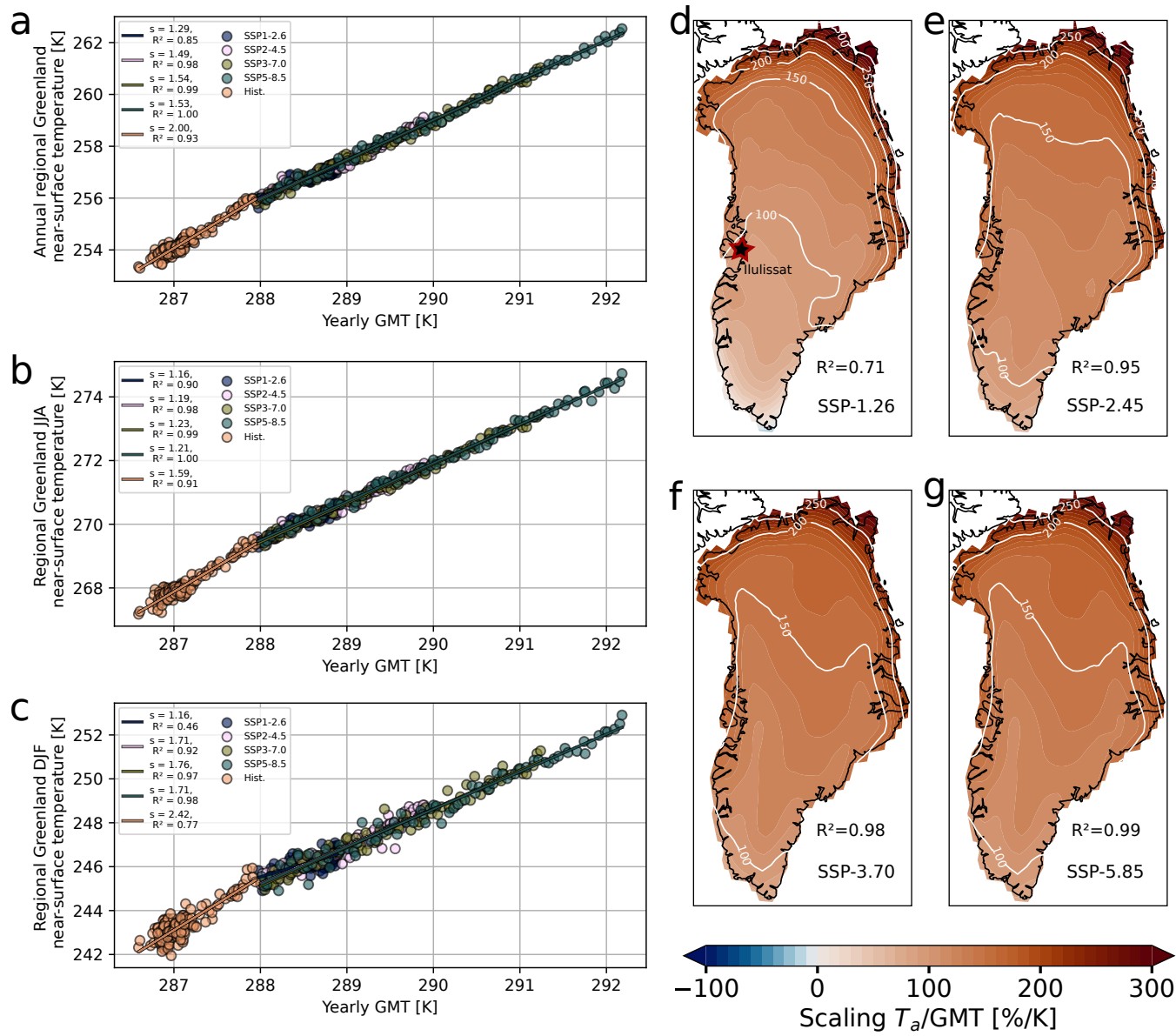

**Figure 2. Ensemble mean and spatially varying scaling factors for near-surface temperatures relative to global mean temperatures.**
**(a)** Fit of the ensemble mean of the annual regional near-surface temperature in Greenland against the ensemble mean GMT for all SSP scenarios and the historical time period. **(b,c,)** Same as **a** but for the summer (JJA) and winter (DJF) temperatures in Greenland, respectively. The DJF temperatures generally increase faster than the JJA temperatures for all scenarios except for SSP1-2.6. **(d)** Regional scaling factors for the ensemble mean of annual surface temperatures in Greenland against GMT for SSP1-2.6. The contour at 100% delineates the area where the regional near-surface temperature increases faster than the GMT. The Ilulissat Icefjord is denoted by the star. **(e, f, g)** Same as **d** but for SSP2-4.5, SSP3-7.0 and SSP5-8.5, respectively. In all scenarios, northern Greenland warms fastest compared to the GMT, with a gradient towards the South. The Southern tip of the GrIS warms slower than the GMT in each scenario.

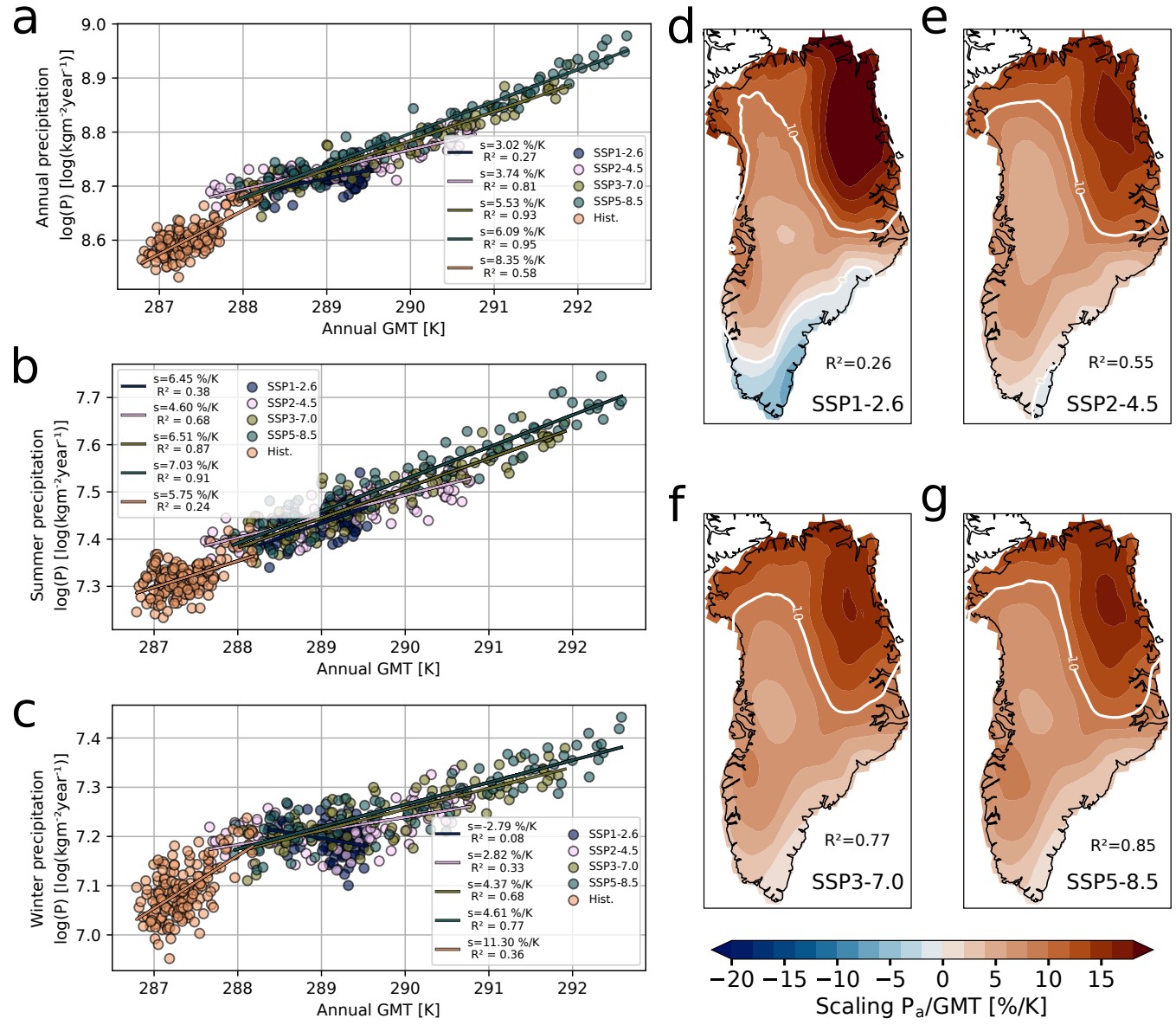

**Figure 3. Ensemble mean and spatial varying scaling factors for precipitation rates relative to global mean temperatures.** **(a)** Fit of the ensemble mean of the annual precipitation rates in Greenland against the ensemble mean GMT for all SSP scenarios and the historical time period. **(b,c,)** Same as **a** but for the summer (JJA) and winter (DJF) precipitation rates in Greenland, respectively. The JJA precipitation rates generally increase faster than the DJF precipitation for all scenarios except for the historical time period. **(d)** Regional scaling factors for the ensemble mean of annual precipitation rates in Greenland against GMT for SSP1-2.6. The white contour at 0%/K denotes the area where the regional precipitation rates decrease. **(e, f, g)** Same as **d** but for SSP2-4.5, SSP3-7.0 and SSP5-8.5, respectively. In all scenarios, precipitation rates increase most strongly in north-eastern Greenland, with a north-south and east-west gradient.

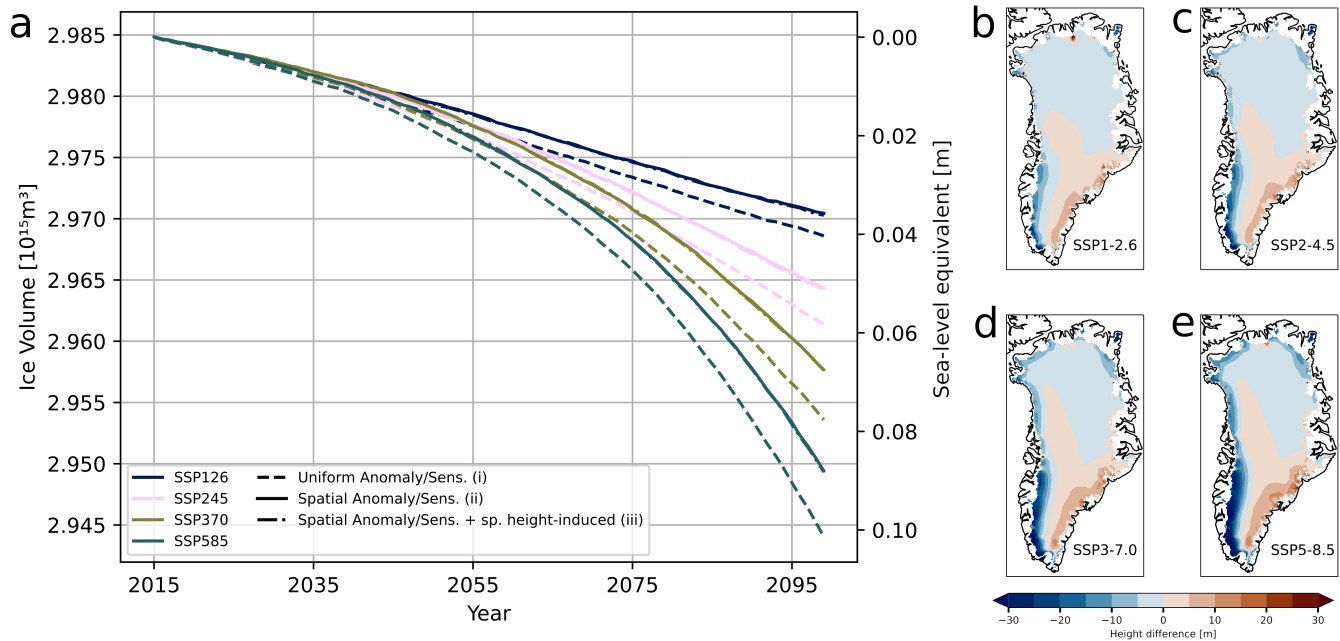

**Figure 4. Comparison of simulated short-term ice volume change between uniform and spatially resolved near-surface temperature anomalies and precipitation sensitivities.** **(a)** Simulated ice volume change from 2015 until 2100. PISM is forced with the ensemble mean of the near-surface temperature anomalies and precipitation sensitivities derived from CMIP6. Solid lines denote the ice volume for spatially resolved anomalies, while the dashed lines denote a spatially uniform anomaly. In each case, the spatially resolved anomalies lead to less ice loss than the uniform anomalies. **(b,c,d,e)** Height difference in the ice thickness in the year 2100 between simulations with spatially resolved and uniform near-surface temperature and precipitation anomalies/sensitivities. Blue areas denote regions where the simulated ice thickness is smaller for the uniform anomalies than for the spatially resolved anomalies. Red areas denote regions where the ice is thicker in the uniform anomaly case. Especially on the southwestern margin of the GrIS, the ice is thinner in the uniform anomaly case in the year 2100. The differences are most pronounced in the SSP5-8.5 scenario.

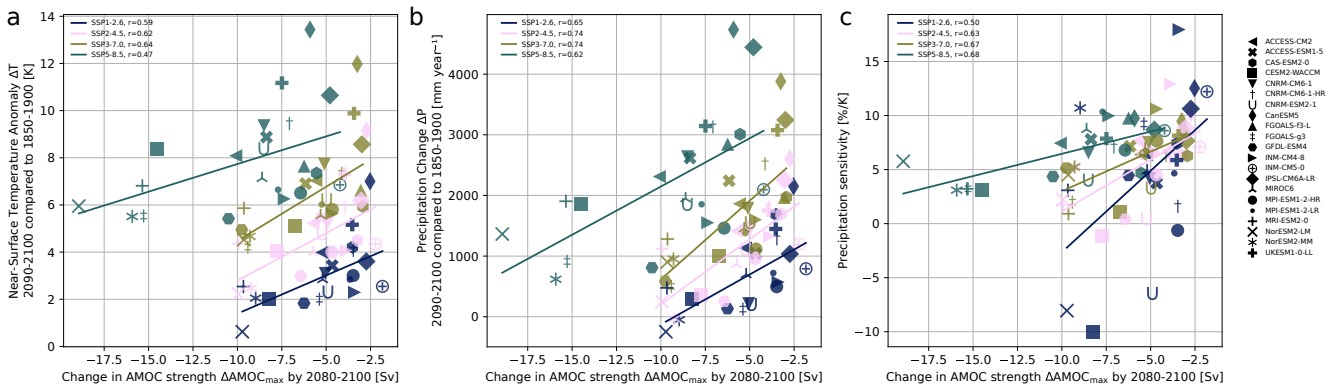

**Figure 5. Relationship between temperature, precipitation and change in AMOC strength. (a)** Change in spatially averaged annual near-surface temperatures by 2090-2100 compared to historical period (1850-1900) in relation to the change of the AMOC strength $AMOC_{max}$ by 2100 (Baker et al., 2023). All scenarios and for all available models are plotted (19 to 21 out of 32 models, dependent on the scenario). There is a moderate relationship between the near-surface temperature anomalies and the change in the AMOC strength. **(b,c)** Same as **a** but for the precipitation anomalies and precipitation sensitivities. The respective $r$-values of the fits are denoted in the legend. AMOC data taken from Baker et al. (2023).

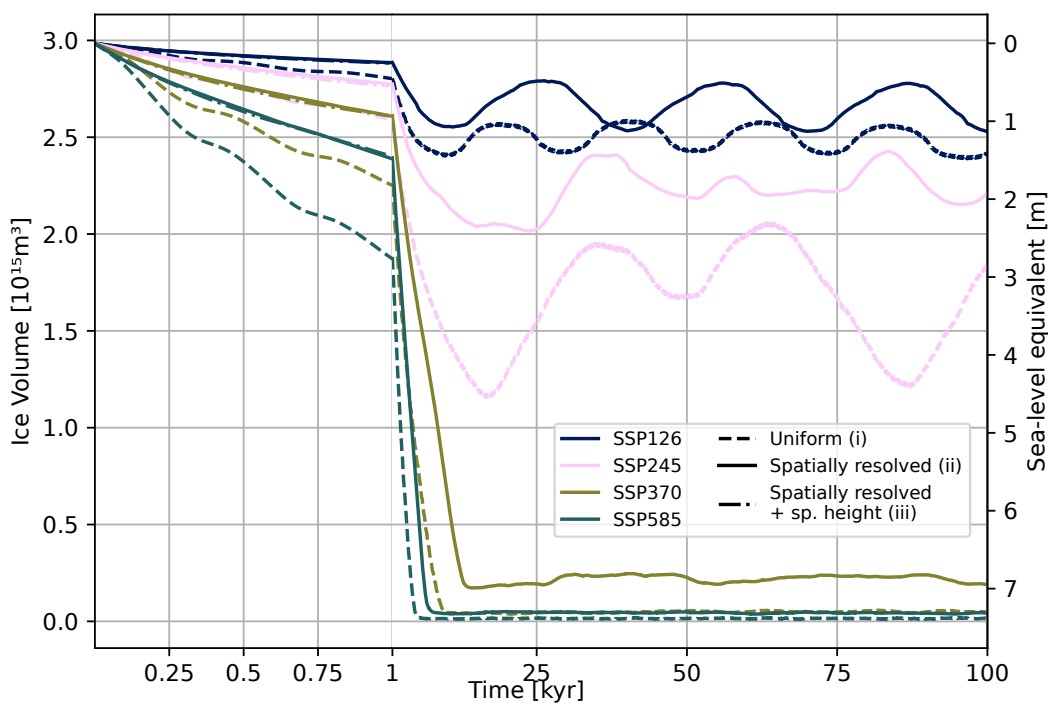

**Figure 6. Simulated long-term ice volume for uniform and spatially resolved anomalies.** Simulated ice volume with PISM for 100 ka. PISM is forced with the ensemble mean of the near-surface temperature anomalies and precipitation sensitivities derived from CMIP6 for the years 2090-2100. Dashed lines denote a uniform anomaly and sensitivity (i), solid lines denote the ice volume for spatially resolved anomalies (ii) and dash-dotted lines (on top of the dashed lines) denote spatially resolved anomalies and sensitivities including spatially resolved height-change-induced precipitation sensitivity (iii). In each case, the spatially resolved anomalies lead to less ice loss than the uniform anomalies. The experiment (iii) is only run for 1,000 years. Oscillations of the ice volume on decamillennial scales are visible.

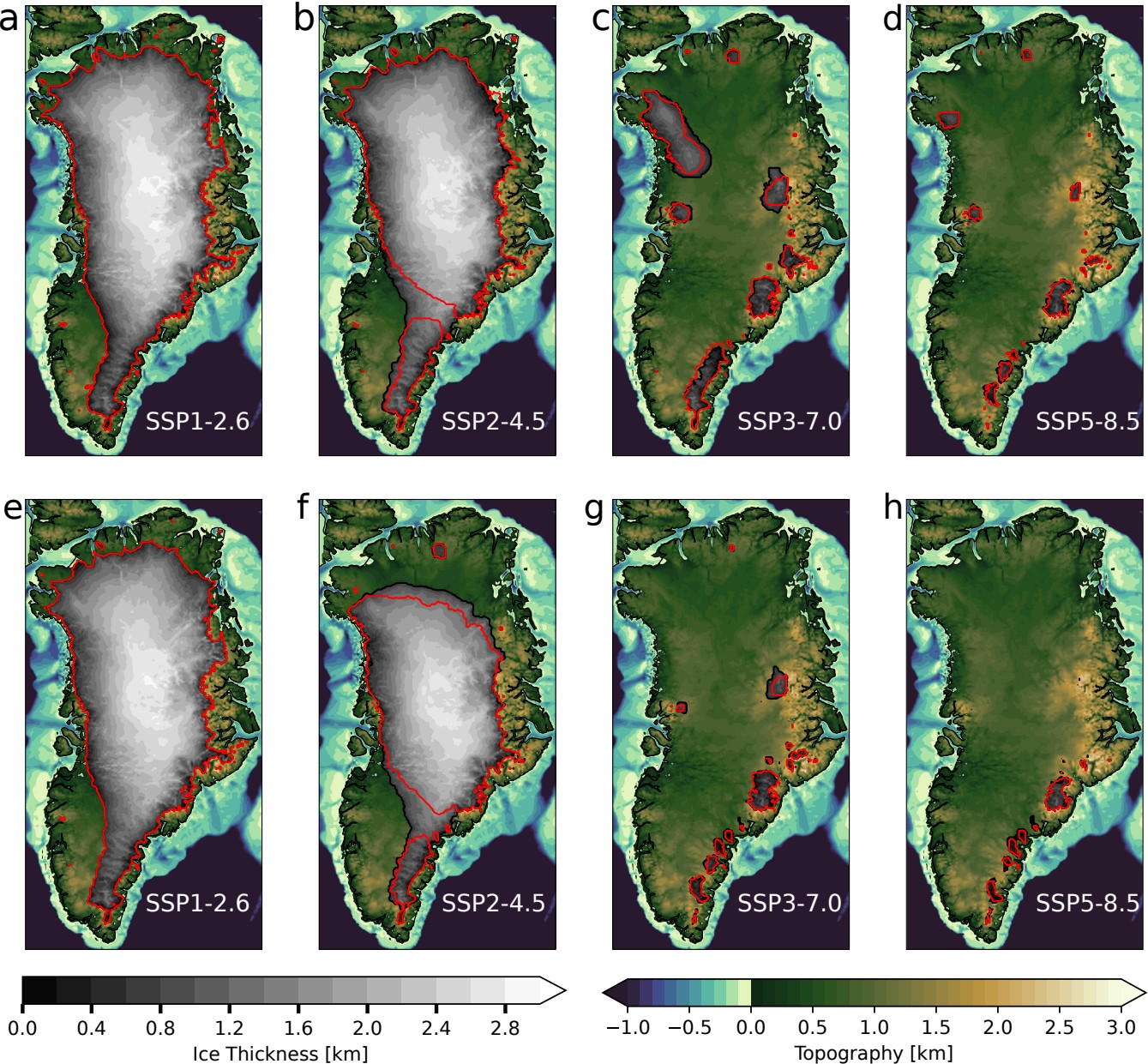

**Figure 7. Maps of simulated long-term ice volume for uniform and spatially resolved near-surface temperature anomalies and precipitation sensitivities.** (a) Simulated ice volume after 100 ka for the SSP1-2.6 scenario and spatially resolved anomalies. The black outline denotes the ice margin at the end of the simulation, while the red line denotes the ice margin of the minimum ice volume within the 100 ka of simulation. (b, c, d) Same as **a** but for the SSP2-4.5, SSP3-7.0 and SSP5-8.5 scenario, respectively. (e, f, g, h) Same as **a, b, c, d** but for uniform anomalies. The minimum ice volume and the ice volume at the end of the simulation is generally smaller than in the case of spatially resolved anomalies. The differences are most pronounced for the SSP2-4.5 scenario, while for the more extreme scenarios the ice sheet is completely lost in both cases.

**Table 1. Overview of experiments.** Each row corresponds to one set of experiments. We run three different sets of experiments, where we vary between (scenario-dependent) spatially uniform and spatially varying precipitation sensitivities, temperature anomalies and height-change induced precipitation sensitivities. The experiments with spatially varying height-change induced precipitation sensitivities are only run for a maximum of 1,000 years. All experiments are run for all emission scenarios.

| Experiment Name | Simulation length [yr] | Precipitation sensitivities | Temperature anomaly | Lapse-Rate [K/km] | Height-change induced precipitation sensitivities [%/K] |
|---|---|---|---|---|---|
| Uniform (i) | 85 & 100,000 | uniform | uniform | uniform [0, 5-8] | uniform [0-7] |
| Spatially Variable (ii) | 85 & 100,000 | spatially varying | spatially varying | uniform [0, 5-8] | uniform [0-7] |
| Spatially Variable + sp. height-induced (iii) | 85 & 1,000 | spatially varying | spatially varying | uniform [6] | spatially varying |

*Code and data availability.* The CMIP6 data is freely available at https://aims2.llnl.gov/search/cmip6/. The regridded data for Greenland as well as the anomalies and sensitivities are available on Zenodo at https://doi.org/10.5281/zenodo.11378715. The source code for the ice sheet model PISM is freely available at https://github.com/pism/pism. Scripts used for the analysis are available from the corresponding author upon request.

# Appendix A

## A1 Temperatures in CMIP6

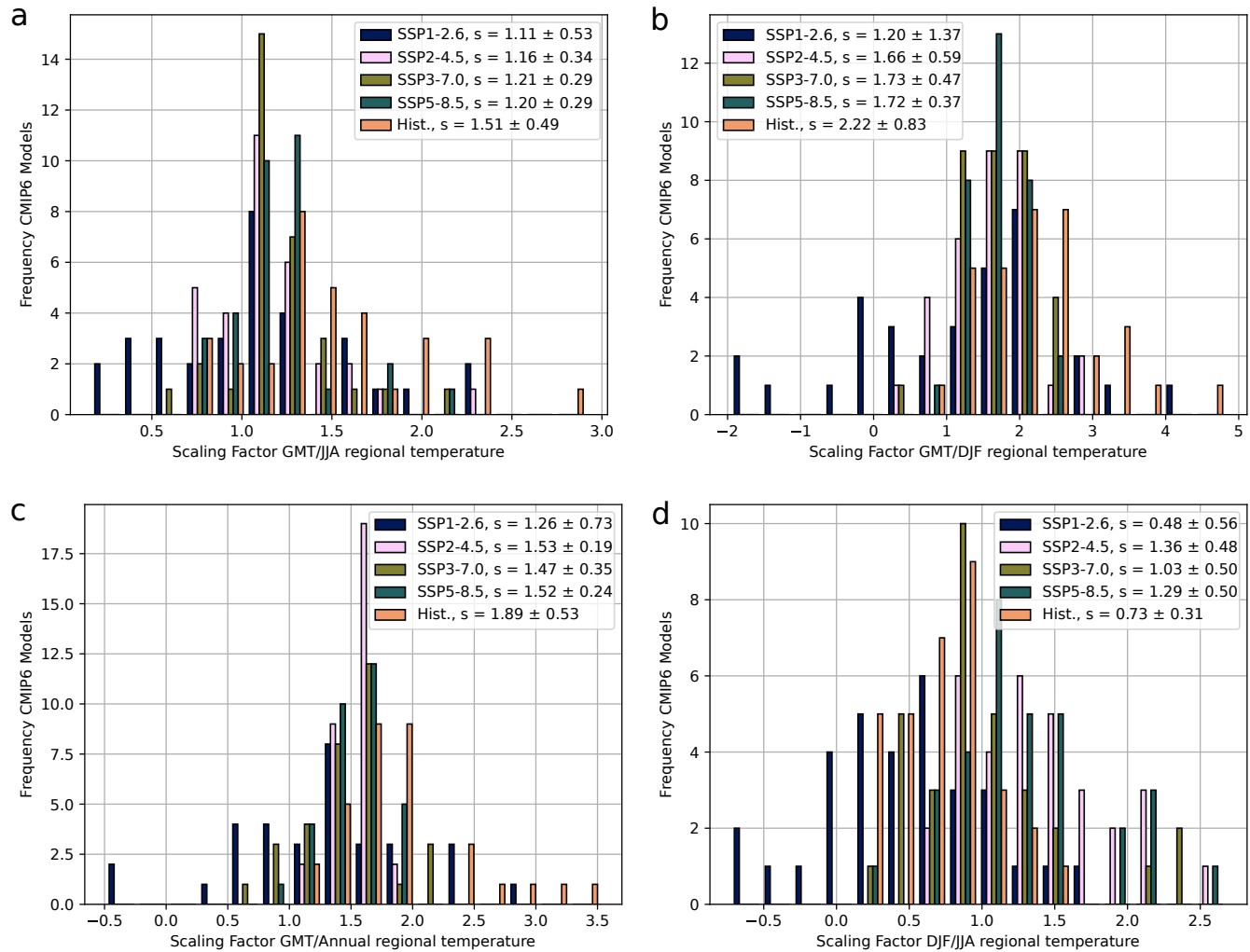

**Figure A1. Histograms of seasonal and annual near-surface temperature scaling factors against GMT for all scenarios and models.**
(a) Scaling factors between spatially averaged summer near-surface temperature (JJA) in Greenland and GMT derived from a linear fit for each model and all scenarios. The ensemble mean and standard deviation for each scenario are given in the legend. (b) Same as **a** but for winter temperatures (DJF). The model spread is larger than for summer scaling factors, especially for the SSP1-2.6 scenario. In all cases, the winter near-surface temperature warms faster than the summer temperatures. (c) Same as **a** but for mean annual temperatures in Greenland. (d) Same as **a** but for scaling factors between regional summer and winter temperatures in Greenland. There is considerable spread in the model response.

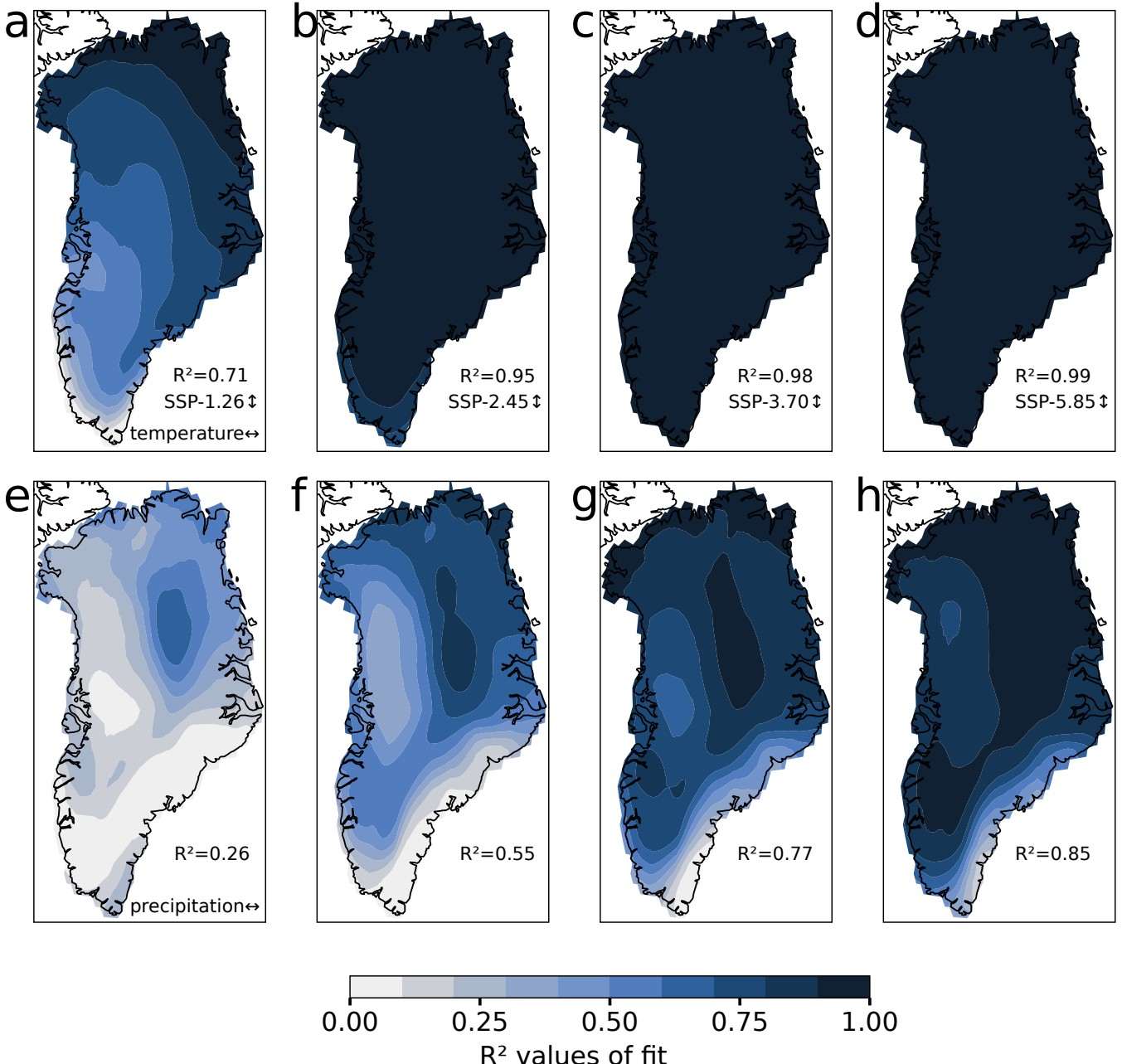

**Figure B1. Maps of $R^2$-values for the linear fit of annual mean temperatures and precipitation rates for all scenarios. (a)** Map of $R^2$-values of linear fit between ensemble mean of regional annual mean near-surface temperature in Greenland and ensemble mean GMT for the SSP1-2.6 scenario. The spatially averaged $R^2$-value is denoted in the right bottom. There is a clear southwest-northeast gradient with southwest Greenland showing the lowest $R^2$-values. **(b,c,d)** Same as **a** but for the SSP2-4.5, SSP3-7.0 and SSP5-8.5 scenarios, respectively. **(e,f,g,h)** Same as **a,b,c,d** but for a fit of the logarithm of ensemble mean precipitation rates against ensemble mean GMT. The ice sheet interior shows the highest $R^2$-values, while the margins, especially the southeastern margin, show low $R^2$-values in all scenarios.

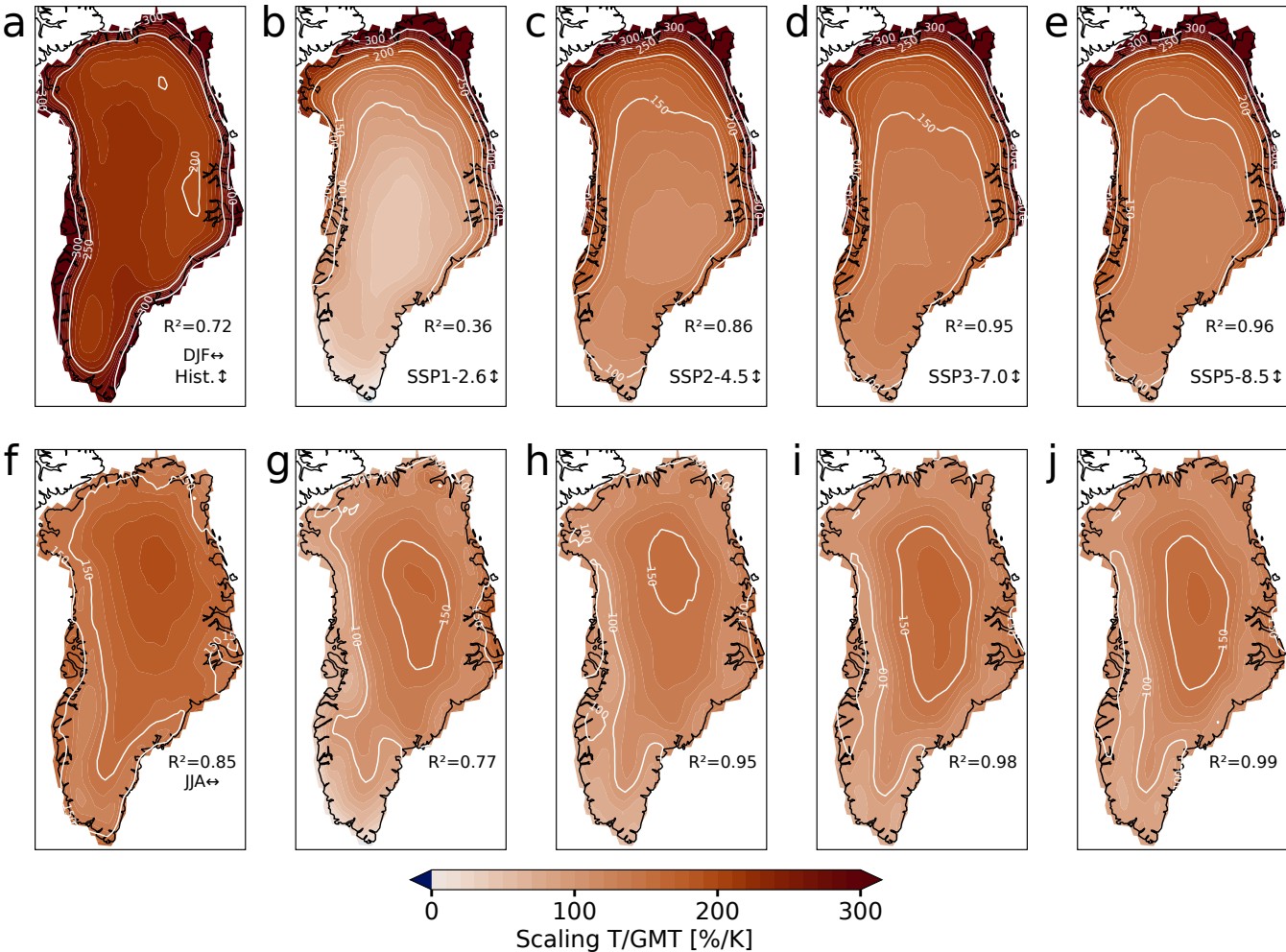

**Figure C1. Maps of scaling factors of ensemble mean temperatures in Greenland for winter (DJF) and summer (JJA) and all scenarios.** **(a)** Scaling factors between ensemble mean winter temperatures and ensemble mean GMT for the historical period (1850-2015). The margins warm faster than the interior of the ice sheet. The spatially weighted mean $R^2$-value is given in the right bottom. **(b,c,d,e)** Same as **a** but for SSP1-2.6, SSP2-4.5, SSP3-7.0 and SSP5-8.5 scenario, respectively. A clear gradient between southern and northern Greenland is visible, with the northern part warming up to 3 times faster than the southern part. **(f,g,h,i,j)** Same as **a,b,c,d,e** but for ensemble mean summer (JJA) temperatures. In all scenarios, the interior of the ice sheet warms faster than the margins.

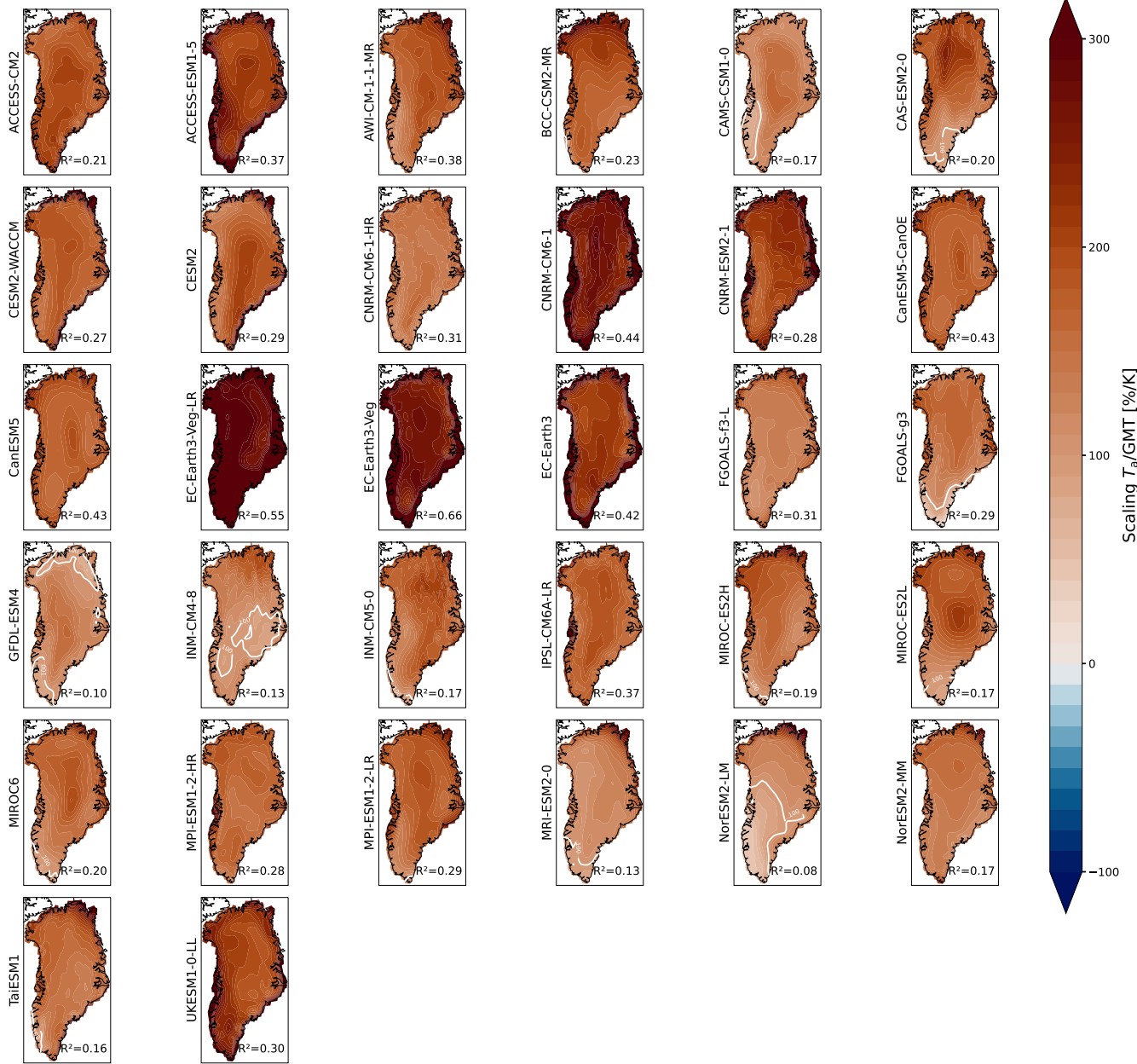

**Figure D1. Maps of annual near-surface temperature scaling factors for each CMIP6 model for the historical period.** Maps of the annual near-surface temperature scaling factors for all models for the historical period (1850-2015). Most models show a faster warming of Greenland compared to the GMT. The white contour denotes the areas that show a slower increase than the GMT. The spatially weighted $R^2$-value is given for each model.

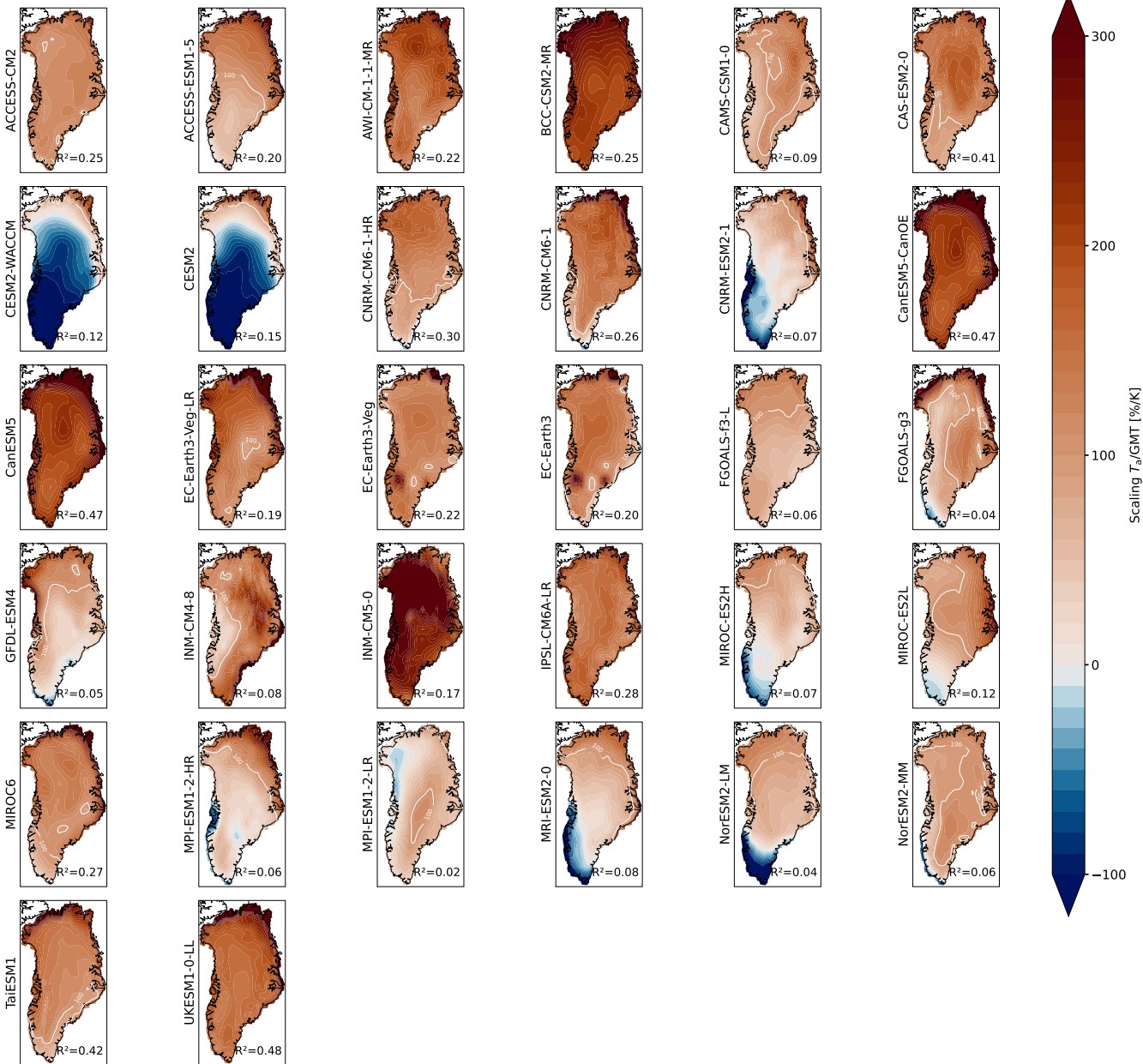

**Figure E1. Maps of annual near-surface temperature scaling factors for each CMIP6 model for the SSP1-2.6 scenario.** Maps of the annual near-surface temperature scaling factors for all models for the SSP1-2.6 scenario (2015-2100). Most models show a slower warming of most parts of Greenland compared to the GMT. Some models even predict a decrease in the annual temperatures in parts of Greenland. The white contour denotes the areas which show a slower increase than the GMT. The spatially weighted $R^2$-value is denoted for each model.

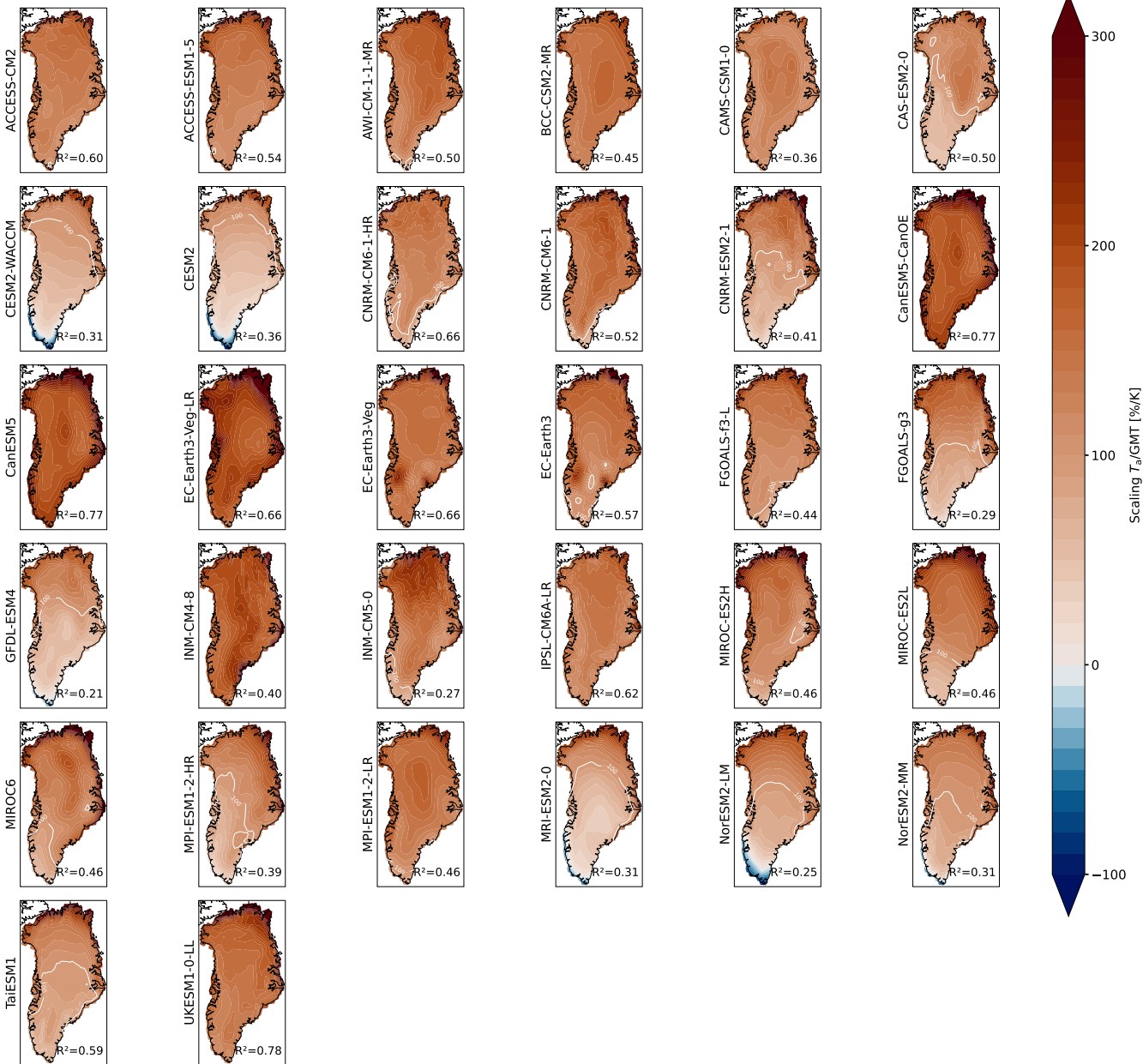

**Figure F1. Maps of annual near-surface temperature scaling factors for each CMIP6 model for the SSP2-4.5 scenario.** Maps of the annual near-surface temperature scaling factors for all models for the SSP2-4.5 scenario (2015-2100). Most models show a faster warming of most parts of Greenland compared to the GMT. The white contour denotes the areas which show a slower increase than the GMT. The spatially weighted $R^2$-value is given for each model.

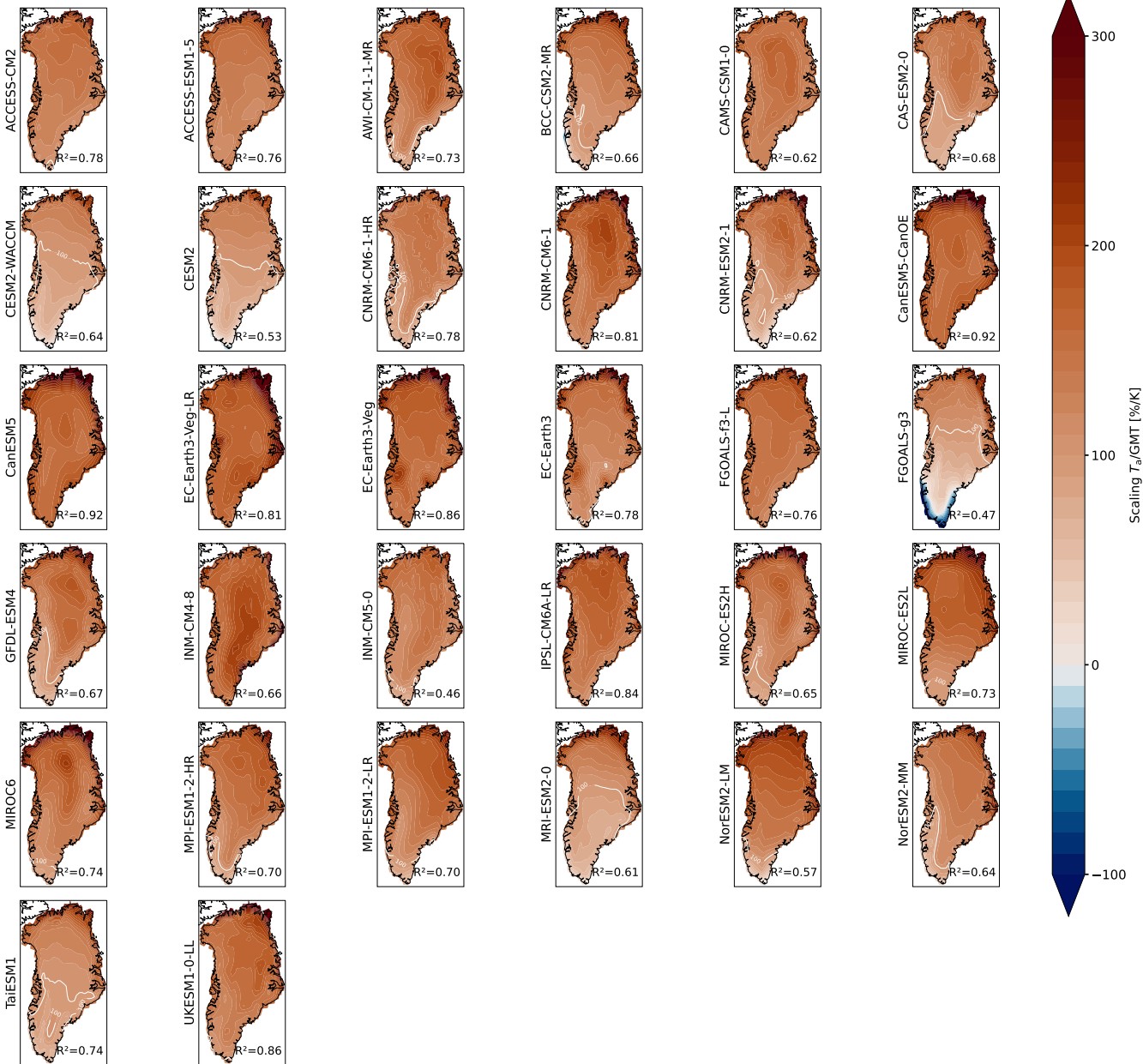

**Figure G1. Maps of annual near-surface temperature scaling factors for each CMIP6 model for the SSP3-7.0 scenario.** Maps of the annual near-surface temperature scaling factors for all models for the SSP3-7.0 scenario (2015-2100). Most models show a faster warming of most parts of Greenland compared to the GMT. The white contour denotes the areas which show a slower increase than the GMT. The spatially weighted $R^2$-value is denoted for each model.

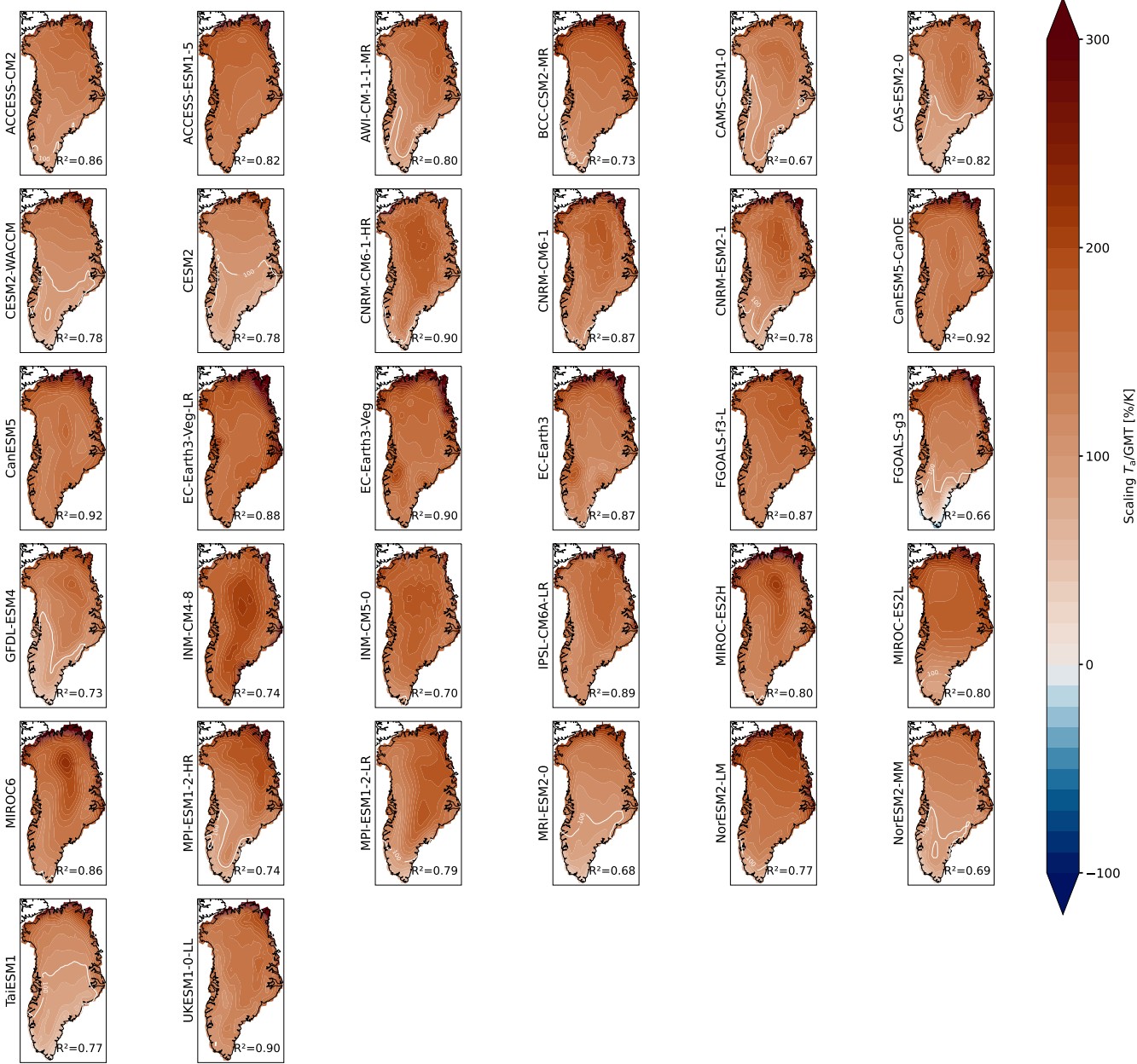

**Figure H1. Maps of annual near-surface temperature scaling factors for each CMIP6 model for the SSP5-8.5 scenario.** Maps of the annual near-surface temperature scaling factors for all models for the SSP5-8.5 scenario (2015-2100). Most models show a faster warming of whole Greenland compared to the GMT. Some models predict a slower warming of southern Greenland compared to the GMT. The white contour denotes the areas which show a slower increase than the GMT. The spatially weighted $R^2$-value is given for each model.

# H1 Precipitation in CMIP6

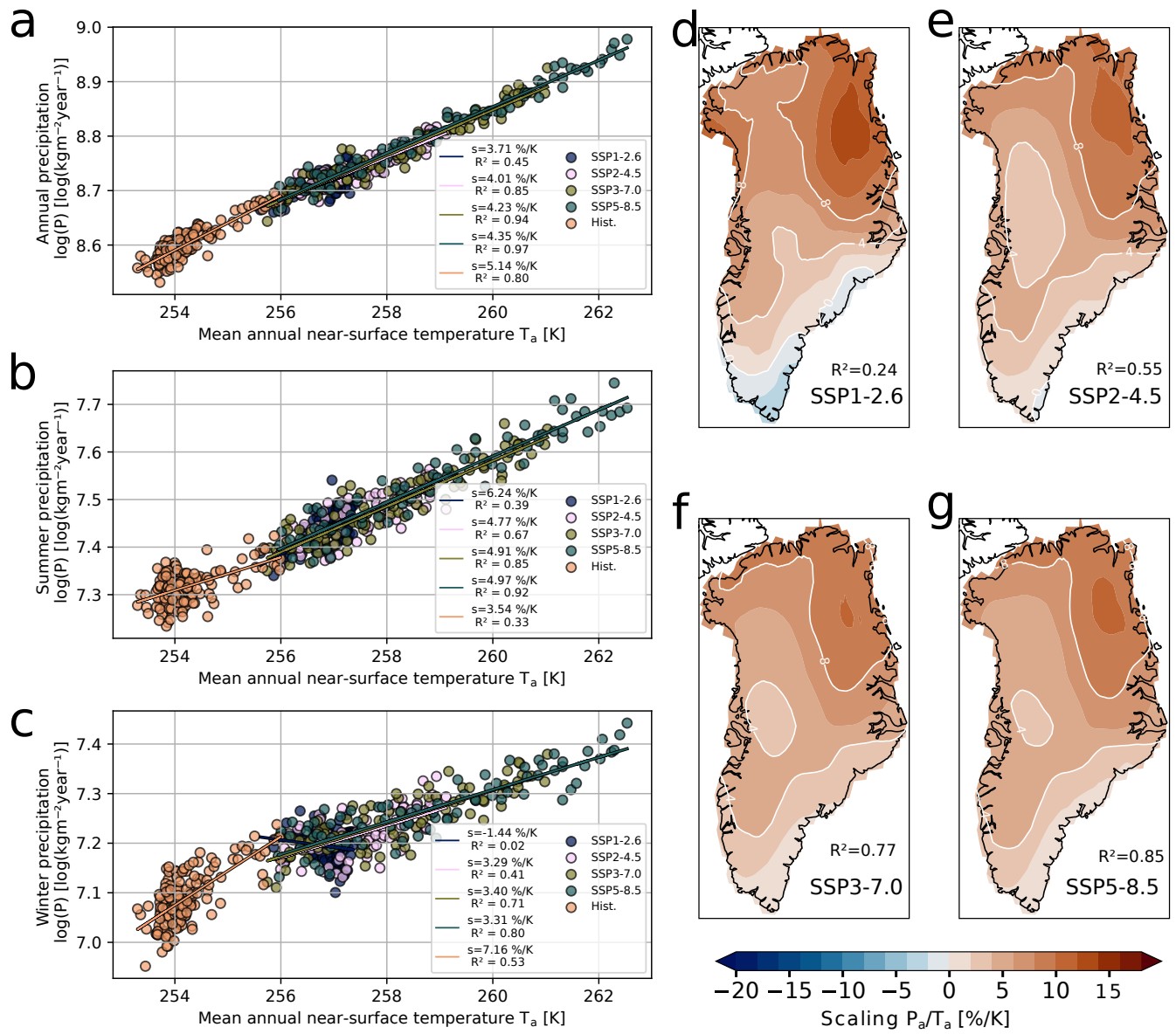

**Figure I1. Ensemble mean and spatially varying scaling factors for precipitation rates relative to annual near-surface temperature in Greenland.** (a) Fit of the ensemble mean of the annual precipitation rates in Greenland against the ensemble mean annual near-surface temperatures for all SSP scenarios and the historical time period. (b,c,) Same as **a** but for the summer (JJA) and winter (DJF) precipitation rates in Greenland, respectively. The JJA precipitation rates generally increase faster than the DJF precipitation for all scenarios except for the historical time period. (d) Regional scaling factors for ensemble mean of annual precipitation rates in Greenland against mean annual near-surface temperatures for SSP1-2.6. The white contour at 0%/K denotes the area where the regional precipitation rates decrease. (e, f, g) Same as **d** but for SSP2-4.5, SSP3-7.0 and SSP5-8.5, respectively. In all scenarios, precipitation rates increase most strongly in north-eastern Greenland, with a north-south and east-west gradient.

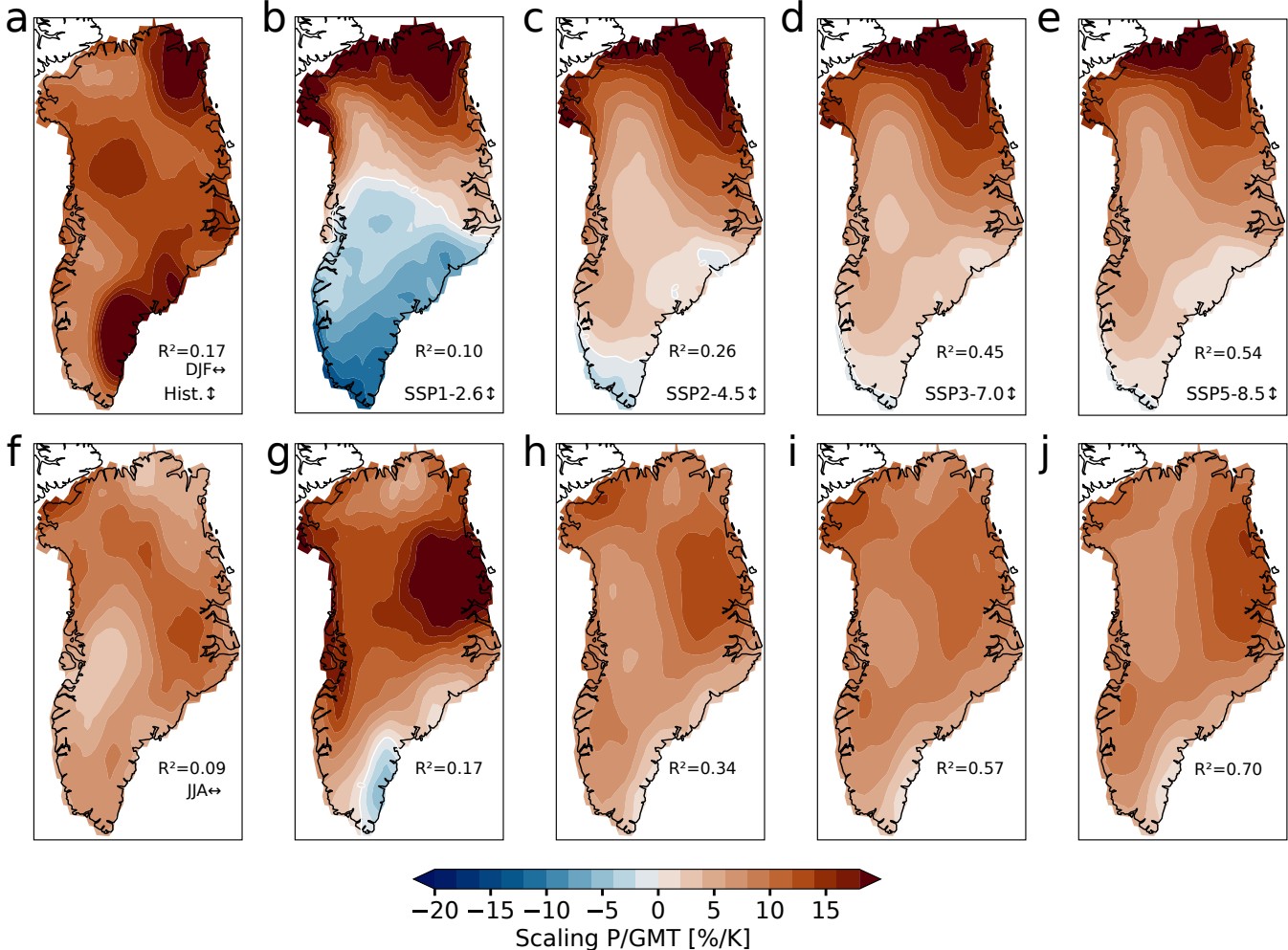

**Figure J1. Maps of precipitation sensitivities of ensemble mean precipitation rates for DJF and JJA and all scenarios. (a)** Precipitation sensitivities of ensemble mean winter precipitation rates and ensemble mean GMT for the historical period (1850-2015). The spatially weighted mean $R^2$-value is given. **(b,c,d,e)** Same as **a** but for SSP1-2.6, SSP2-4.5, SSP3-7.0 and SSP5-8.5 scenario, respectively. A clear gradient between southwestern and northeastern Greenland is visible, with the sensitivities in the northern part exceeding 15%/K. **(f,g,h,i,j)** Same as **a,b,c,d,e** but for ensemble mean summer (JJA) precipitation rates.

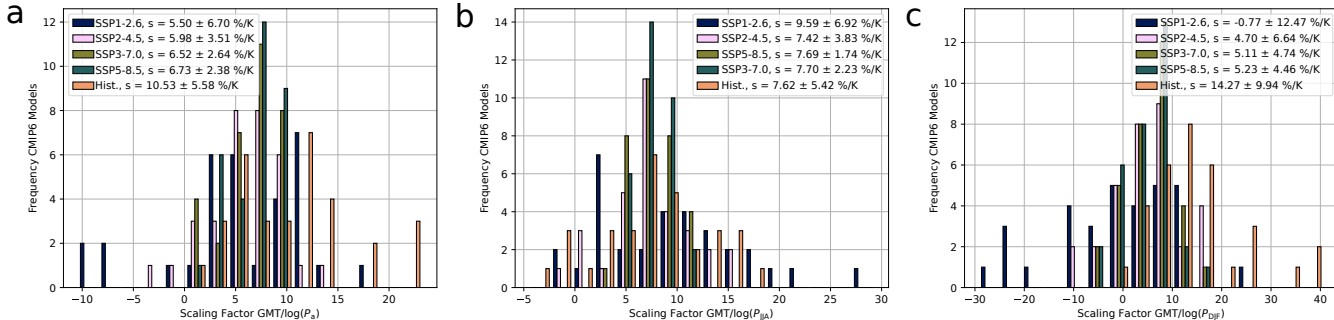

**Figure K1. Histograms of seasonal and annual precipitation sensitivities against GMT for all scenarios and models. (a)** Sensitivities of spatially averaged annual precipitation rates in Greenland and GMT derived from a linear fit of $\ln(P)$ for each model and all scenarios. The ensemble mean and standard deviation for each scenario are given in the legend. There is a considerable spread in the model response. Especially for the SSP1-2.6 scenario, the uncertainty is larger than the mean. **(b)** Same as **a** but for summer (JJA) precipitation sensitivities. **(c)** Same as **a** but for winter (DJF) precipitation sensitivities in Greenland.

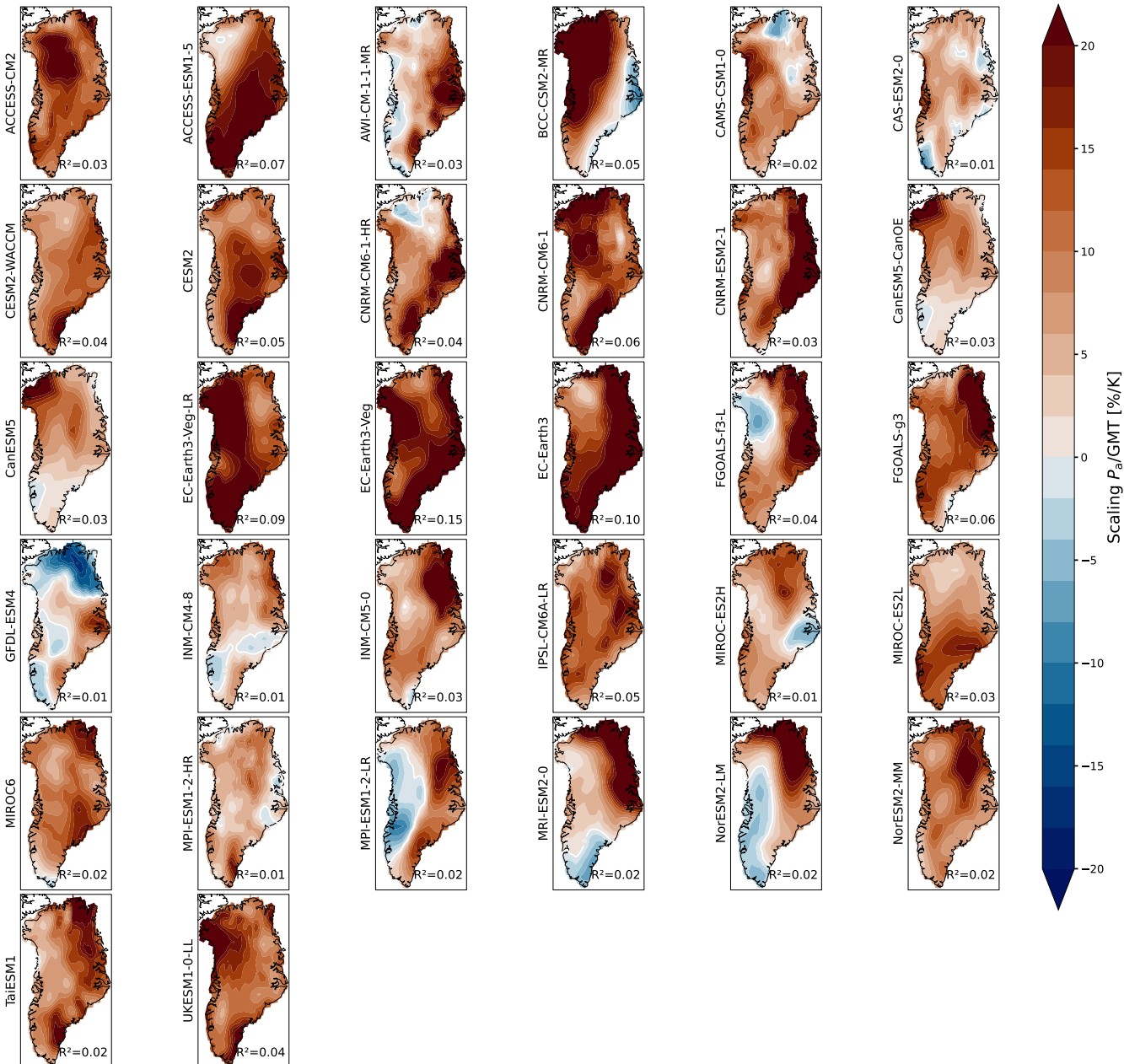

**Figure L1. Maps of annual precipitation sensitivities for each CMIP6 model for the historical period.** Maps of the precipitation sensitivities for all models for the historical time period (1850-2015). There is no clear pattern in the precipitation sensitivities across the models. The spatially weighted mean $R^2$-value of the fit is very small for most models. The white contour denotes the areas which show a negative precipitation sensitivity, i.e. a decrease in the precipitation rates. The spatially weighted $R^2$-value is denoted for each model.

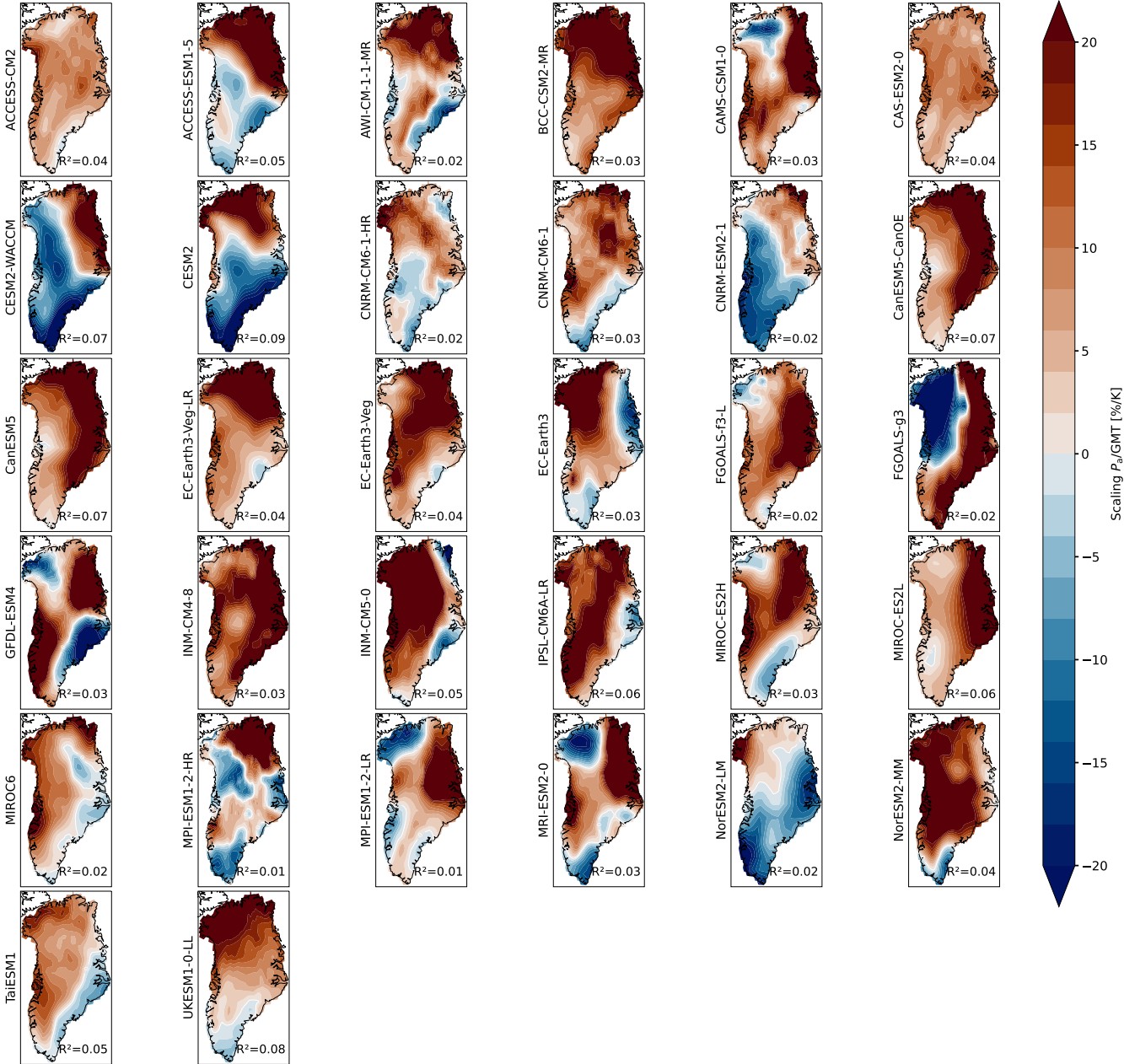

**Figure M1. Maps of annual precipitation sensitivities for each CMIP6 model for the SSP1-2.6 scenario.** Maps of the precipitation sensitivities for all models for the SSP1-2.6 scenario (2015-2100). There is no clear pattern in the precipitation sensitivities across the models. The spatially weighted mean $R^2$-value of the fit is very small for most models. The white contour denotes the areas which show a negative precipitation sensitivity, i.e. a decrease in the precipitation rates. The spatially weighted $R^2$-value is denoted for each model.

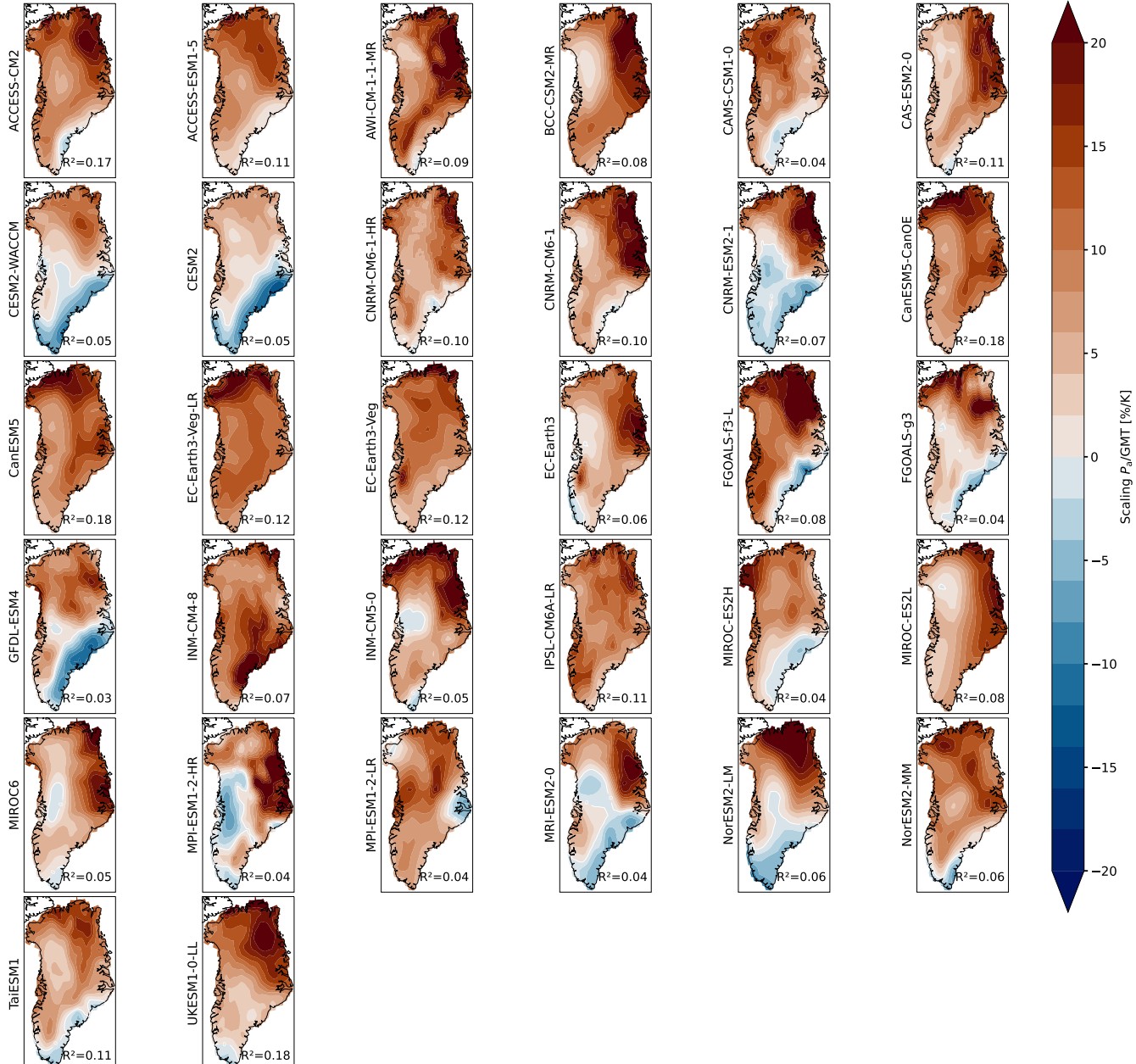

**Figure N1. Maps of annual precipitation sensitivities for each CMIP6 model for the SSP2-4.5 scenario.** Maps of the precipitation sensitivities for all models for the SSP2-4.5 time period (2015-2100). The majority of models show a positive precipitation sensitivity for most parts of Greenland. However, some models show negative sensitivities for the southeastern margin of the GrIS, partially extending into the interior of the ice sheet. However, the spatially weighted mean $R^2$-value of the fit is very small for most models. The white contour denotes the areas which show a negative precipitation sensitivity, i.e. a decrease in the precipitation rates. The spatially weighted $R^2$-value is denoted for each model.

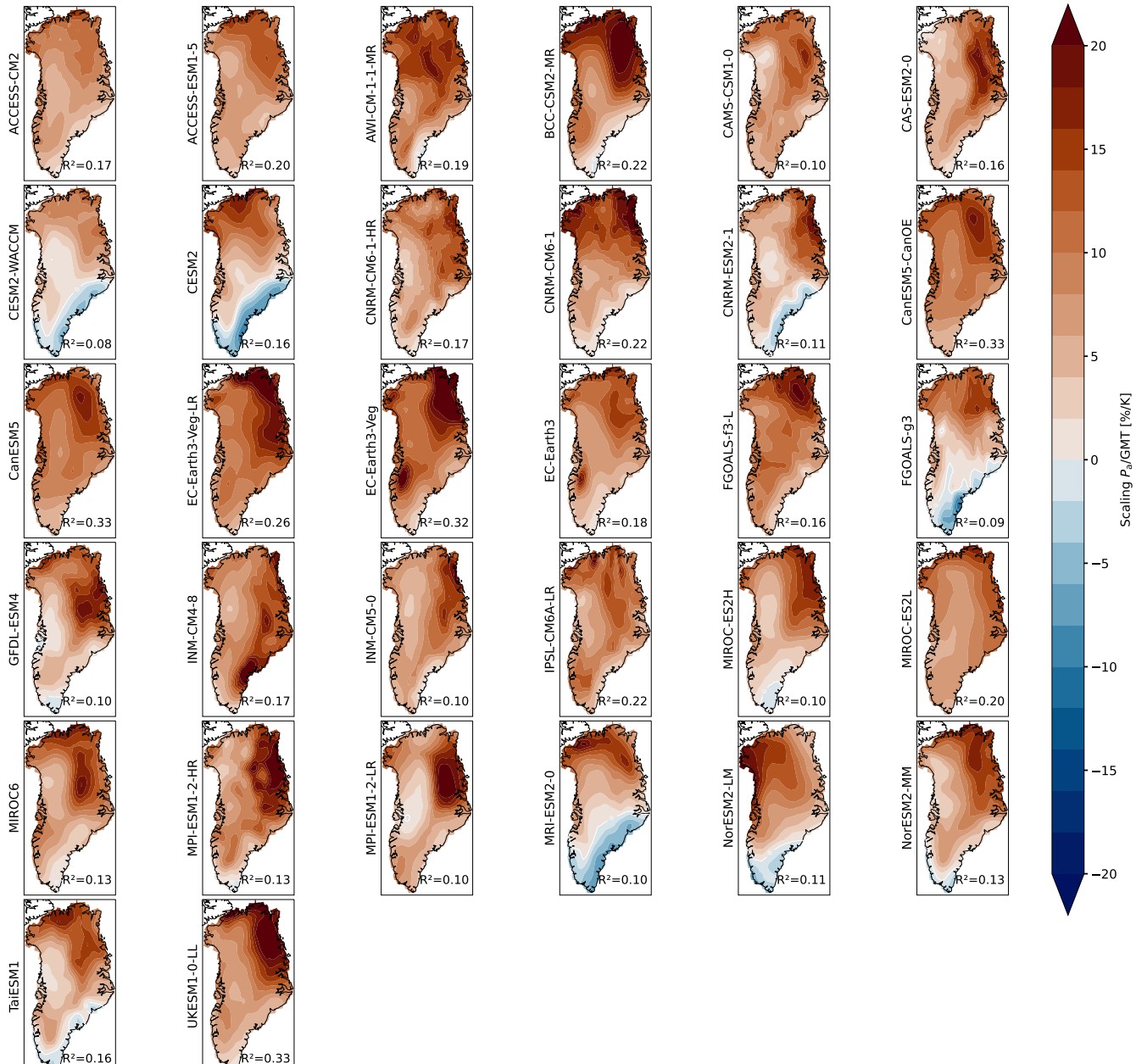

**Figure O1. Maps of annual precipitation sensitivities for each CMIP6 model for the SSP3-7.0 scenario.** Maps of the precipitation sensitivities for all models for the SSP3-7.0 scenario (2015-2100). The majority of models show a positive precipitation sensitivity for most parts of Greenland. However, some models show negative sensitivities for the southeastern margin of the GrIS. The white contour denotes the areas which show a negative precipitation sensitivity, i.e. a decrease in the precipitation rates. The spatially weighted $R^2$-value is given for each model.

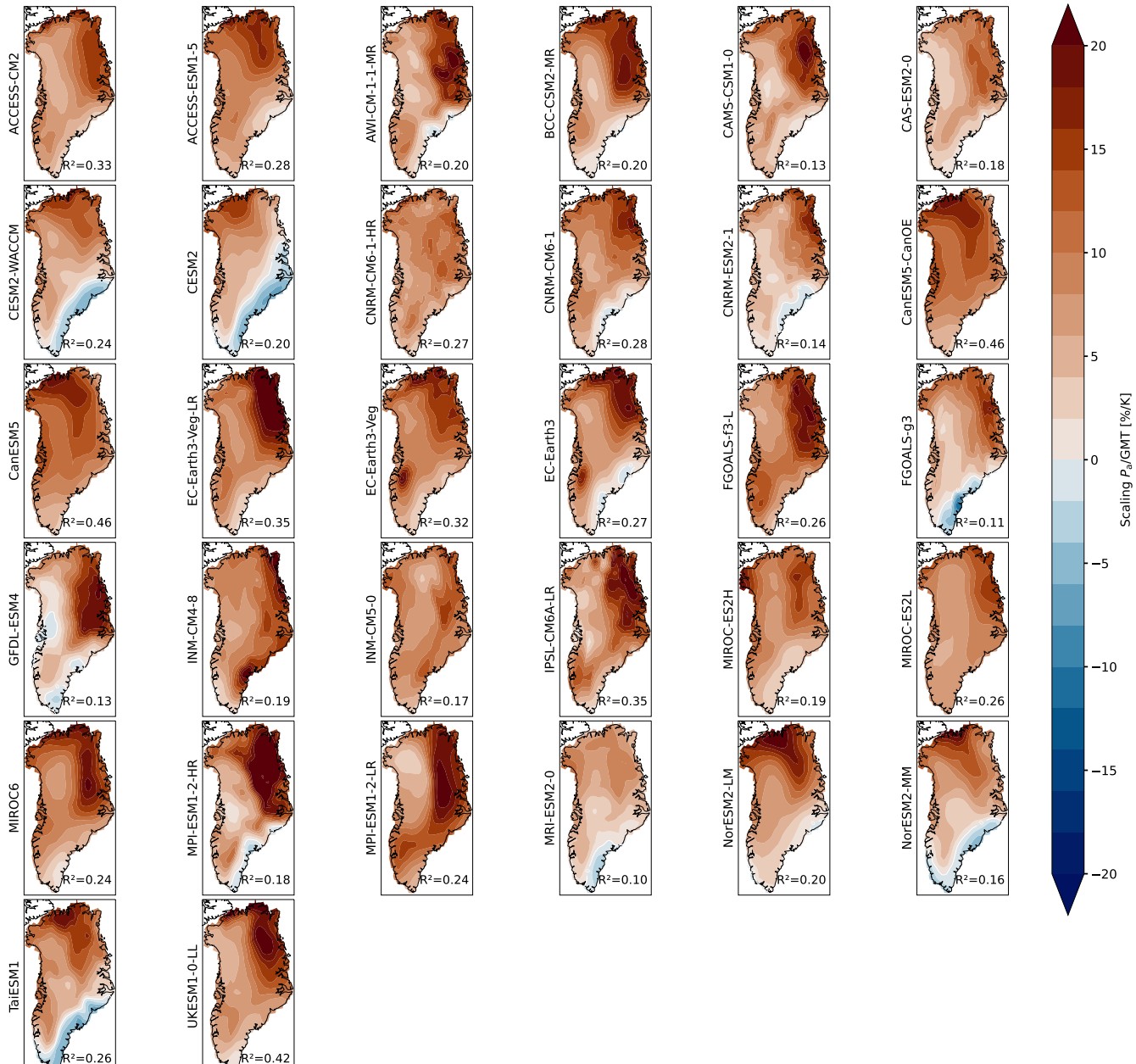

**Figure P1. Maps of annual precipitation sensitivities for each CMIP6 model for the SSP5-8.5 scenario.** Maps of the precipitation sensitivities for all models for the SSP5-8.5 scenario (2015-2100). The majority of models show a positive precipitation sensitivity for most parts of Greenland. However, some models show negative sensitivities for the southeastern margin of the GrIS. The white contour denotes the areas which show a negative precipitation sensitivity, i.e. a decrease in the precipitation rates. The spatially weighted $R^2$-value is given for each model.

**P1    Various**

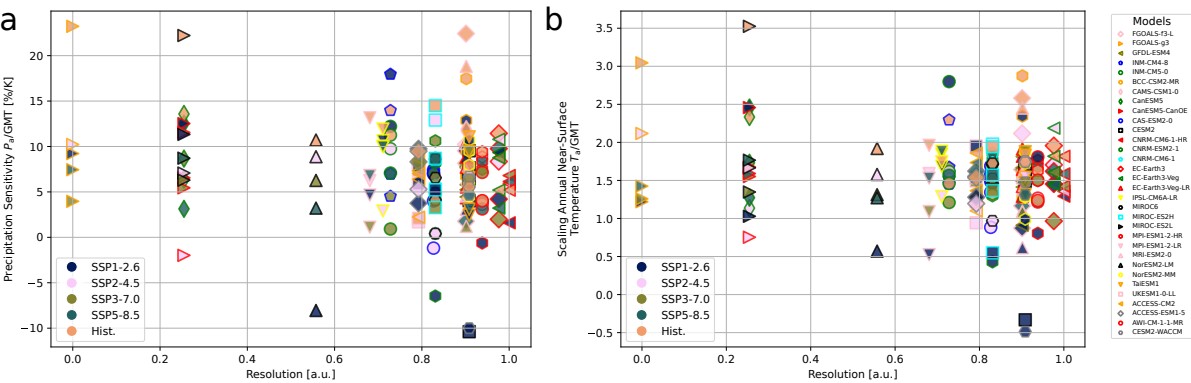

**Figure Q1. Dependency of precipitation sensitivity and temperature scaling on model resolution. (a)**. Spatially averaged precipitation sensitivity against model resolution in arbitrary units, where 0 is the lowest resolution and 1 is the highest resolution. All models for all scenarios are plotted. **(b)** Same as **a** but for the temperature scaling. While the spread in the model response seems to decrease slightly with increasing model resolution, no strong relationship between the scaling/sensitivity and the model resolution is visible.

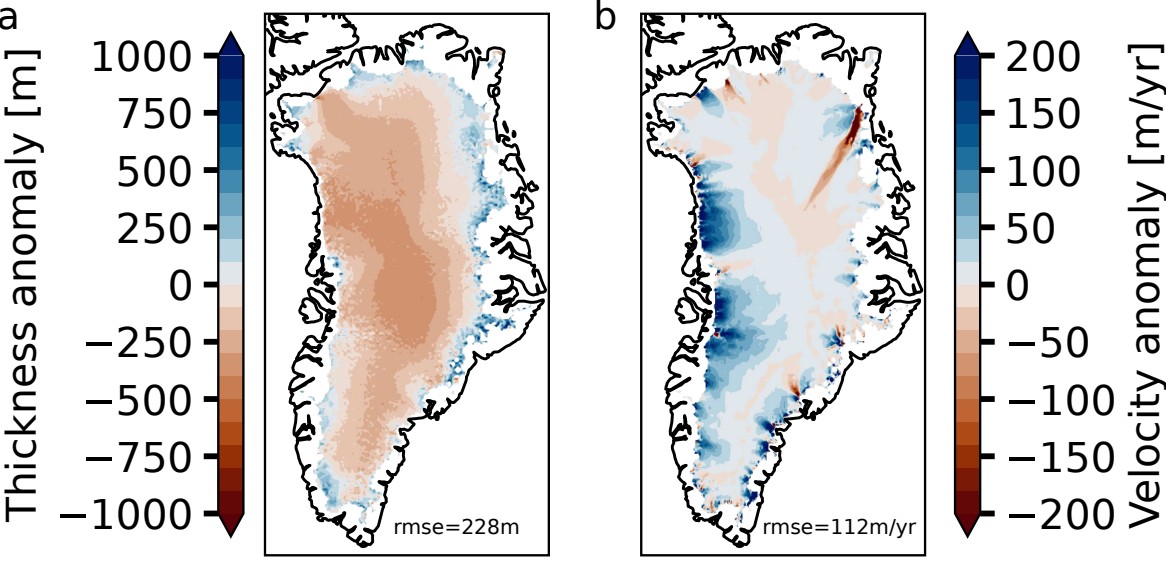

**Figure R1. Difference between initial state and observed ice thickness and velocity. (a)** Thickness anomaly of initial state compared to observational data (Morlighem et al., 2017; Morlighem, 2022). The root mean squared error is 228 m with a general understimation of the thickness in the interior of the ice sheet. **(b)** Same as **a** but for the ice flow velocity (Joughin et al., 2016, 2018). The root mean squared error is 112 m/yr, mostly due to an overestimation of velocities in the western part of the ice sheet.

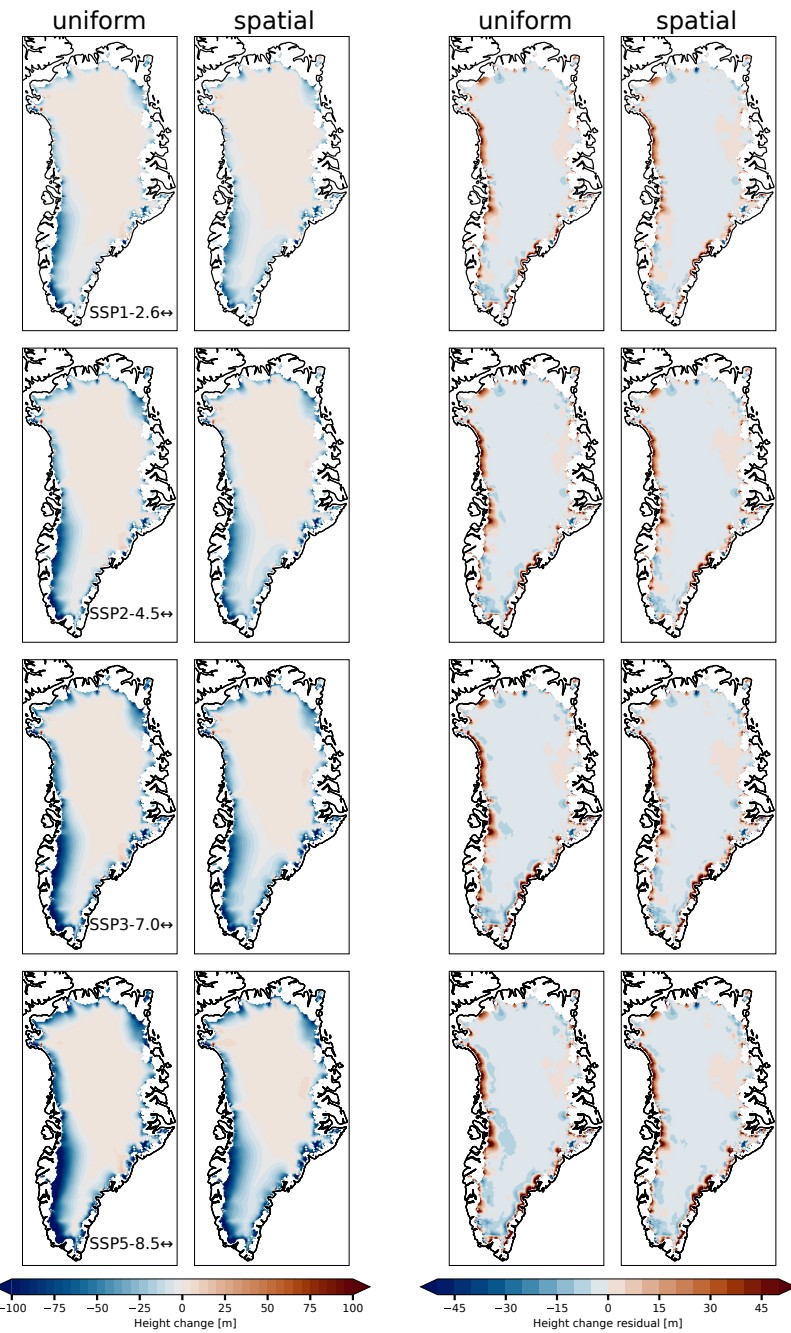

**Figure S1. Maps of absolute height change and dynamic contribution of the height change for the short-term simulations. (left)** The absolute height change after 85 years for each scenario compared to the initial state is plotted for experimental setups (i) and (ii). In each scenario southwestern Greenland shows the biggest height difference. **(right)** Dynamic contribution to the height change, i.e., the height changes that cannot be attributed to changes in the SMB.

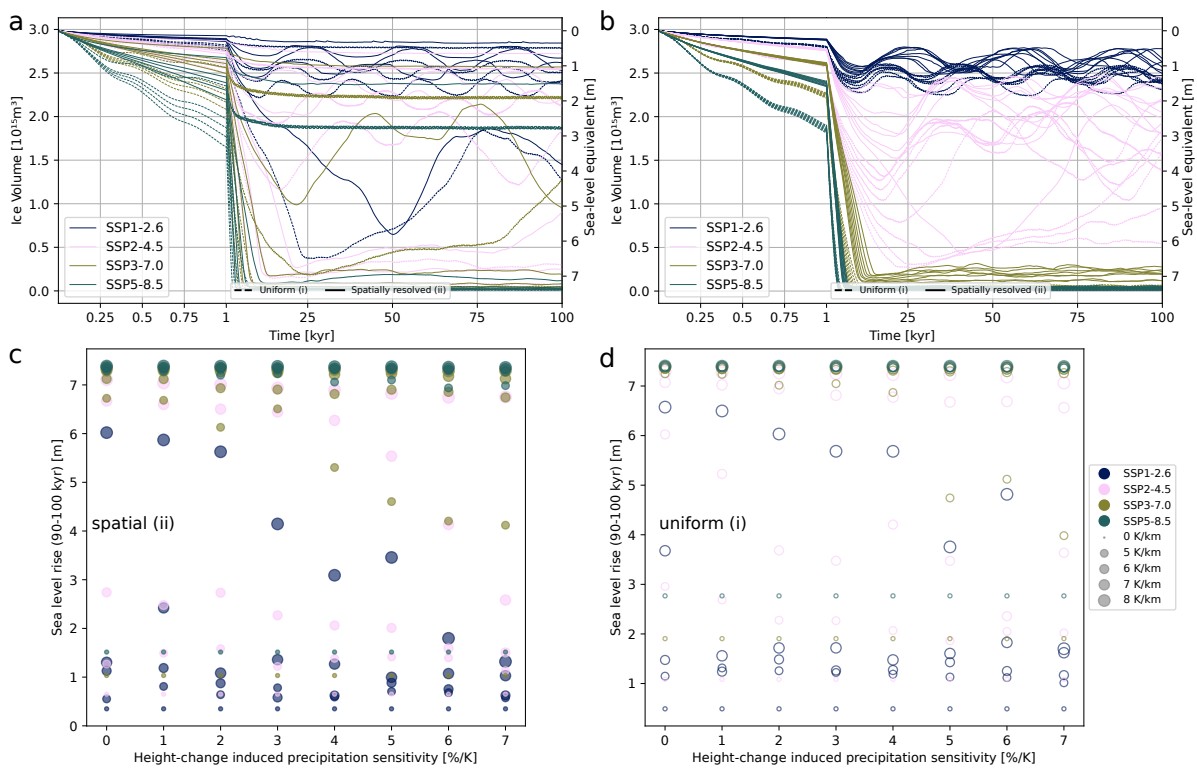

**Figure T1. Sensitivity analysis of long-term simulations for different lapse rates and height-change-induced precipitation sensitivities.**
**(a)** Time series of ice sheet volume for different lapse rate values (0,5,6,7,8 K/km) and a fixed height-change-induced precipitation sensitivity of 5 %/K for all emission scenarios. Dashed lines correspond to experimental setup (i), while solid lines correspond to experimental setup (ii). **(a)** Same as **a** but for fixed lapse-rate (6 K/km) and varying height-change induced precipitation sensitivity (0,1,2,3,4,5,6,7 %/K). Generally, the lapse-rate has a stronger influence on the long-term behaviour than the height-change induced precipitation sensitivity. **(c)** Scatter plot of mean sea-level rise after 90-100 kyr for all combinations of lapse-rates and height-change induced precipitation sensitivities for spatially varying anomalies and sensitivites (ii). The size of the marker corresponds to the lapse-rate value, while the colour corresponds to the emission scenario. **(d)** Same as **c** but for uniform anomalies and sensitivities. In both cases, a clear dependency of the sea-level rise on the lapse-rate and the height-change induced precipitation sensitivity is visible.

*Author contributions.* N.Boc. conceived the study. N.Boc. and N.Boe. designed the study. A.P. assembled the CMIP6 data. N.Bow. and A.P. pre-processed the CMIP6 data. N.Boc. did all the analysis and simulations. All authors discussed the results. N.Boc. wrote the manuscript with contributions from all authors.

*Competing interests.* The authors declare that they have no conflict of interest.

*Acknowledgements.* This work was supported by the UiT Aurora Centre Program, UiT - The Arctic University of Norway (2020), and the Research Council of Norway (project number 314570). This is ClimTip contribution #8; the ClimTip project has received funding from the European Union's Horizon Europe research and innovation programme under grant agreement No. 101137601. N.Boe. acknowledges further funding from the Volkswagen Foundation and the European Union's Horizon 2020 research and innovation program under the Marie Sklodowska-Curie grant agreement No. 956170. Parts of the computations were performed on resources provided by Sigma2 - the National Infrastructure for High Performance Computing and Data Storage in Norway under the project nn8008k and nn9348k. Development of PISM is supported by NASA grants 20-CRYO2020-0052 and 80NSSC22K0274 and NSF grant OAC-2118285. The colormaps for the plots are taken from Crameri et al. (2020). We thank Jonathan Baker for providing the AMOC data.

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
