# Peer review of "Projections of Precipitation and Temperatures in Greenland and the Impact of Spatially Uniform Anomalies on the Evolution of the Ice Sheet"

_EGUsphere, 2024_

## Referee Comment (RC1)

Review of Bochow et al. 2024

**General comment:**

In this paper, the authors provide insights into the importance of considering spatial variations in temperature and precipitation anomalies for accurate modeling of the Greenland ice sheet's response to climate change, highlighting the potential impact of modeling choices on long-term ice sheet stability and sea level rise projections. To do so, Bochow et al. analyzed CMIP6 temperature and precipitation changes in Greenland, deriving spatially resolved and scalar scaling factors for near-surface temperatures and precipitation against global mean temperature and time. The authors then used the Parallel Ice Sheet Model (PISM) to compare the impact of spatially uniform versus spatially resolved anomalies on the Greenland ice sheet's short and long-term behavior.
I think this is an interesting paper, worthy of publication for The Cryosphere. I do have though a couple of major comments which I think the authors should address before publication: one regarding how the methodology is presented, and another one regarding an aspect of the methodology (run-time lapse rate and precipitation correction factors).

**Major comments:**

1) I think that the methodology subsection 2.2 PISM lacks details and is not clear enough. For instance, there are too few details about the initialization procedure and some aspects of the model forcing (calving rates, see specific comments). Also, while I understand in principle how the simulations were forced, I found the text explaining the forcing not entirely clear. For instance, it seems to me that applying spatially variable anomalies of temperature and precipitation is not different from any other study using GCM output (including ISMIP6 studies). In this regard, the link with the scaling factors is not entirely clear (see second comment). I suggest expanding the text explaining clearly the different experiments (perhaps including a table for different experiments, or a diagram showing how the forcing is produced).

2) I was a bit disappointed to read that you applied constant lapse rate and elevation-change-induced precipitation scaling throughout the run. Somehow it is contradicting the premises of the manuscript - that is, going beyond values calculated with previous version of CMIP models. I think that using the values you inferred for precipitation correction would be the missing link between the first part of your results (deriving parameters) and the second part (ice sheet model simulations). This would shed light not only on scalar vs spatially varying temperature and precipitation, but also on scalar vs spatially varying precipitation correction. If what I suggest is too complicated, it would be enough to test at

least different scalar values for the precipitation correction (within the range values found in this study). A similar thing could (should) be done for the lapse rate (although the lapse rate is not calculated in this study: perhaps you could use values taken from Feenstra et al., 2024 "Effect of elevation feedbacks and climate mitigation on future Greenland ice sheet melt", The Cryosphere (preprint).

**Specific comments:**

*Abstract*

L2: I think uniform anomalies should go first in the sentence than parametrisation schemes.

L3: Is this paper looking at different model parameters, or precipitation correction only? Suggest rephrasing like 'it is often assumed … based on old generation models'.

L12: I would try to also add here why there is such an overestimation.

1. *Introduction*

L20: surface mass balance is accumulation - ablation; I am not clear if here you also consider ice discharge into the ocean, which seems the case from L23: then I suggest explaining better how mass balance is calculated. Also, missing the reference (probably Otosaka et al. 2023: perhaps you can put together this sentence with the next about global sea level rise).

L25: would explain why (ice margin retreating and losing contact with the ocean).

L27: I would suggest rephrasing the second part, something like: "to ice sheet models coupled to Earth system models of intermediate or full complexity". Or at least you should specify that full complexity ESMs are coupled to a dynamic ice sheet model component - which is not obvious, as most full complexity ESMs are not. Also, I would suggest citing at least ISMIP6 work under the stand-alone ice sheet modelling.
L31: I find 'latest generation' a bit confusing, is it CMIP5? CESM and UKESM are CMIP6 models with ice sheet coupling. Also, I think you could specifically mention the coupling work with a dynamic Greenland ice sheet component.

L38: maybe uniform instead of scalar?

L43: I find the notation L(T) a bit confusing, as it looks like a function of the temperature T - but it is instead a constant (although it depends on the temperature). Maybe L_T?
L51: is 23 °C the annual mean temperature between 1996 and 2019? I am not entirely convinced how this simple calculation is relevant for the paper. Perhaps the authors can expand on that?

L65: Please add citation for CMIP6.

L66: Please add citation for SSPs. Also, SSP abbreviation was already used before introducing it (L75).

**2. *Data and Methods**

L80: Gaussian with capital G?

L83: What did you do when the land-ice fraction variable was not available?

L86: precipitations

L99: Perhaps spatial*ly variable* scaling factors?

L111: I think this choice should be explained.

L117: I think you should give an overview of what has been done here. Also, "mostly" is too vague - please state clearly what has been done differently.

L123: I am a bit confused here. If you are calculating precipitation scaling with temperature, why then use a constant value for the surface-height-induced near-surface temperature change? I am not necessarily saying you should not do that, but I think you should explain this choice.

L125: I suggest adding a few more details about front-retreat calving. It's not clear to me how and why this is implemented, especially for the long-term runs.

L127: I assume what comes between L127 and L132 is about the spin-up. I suggest then moving the last sentence (resolution) upwards, as I imagine it also regards the future projections.

**3. *Greenlands climate in CMIP6**

L135-L140: the beginning of this section feels a bit strange, as the first lines seem to belong either to the introduction or the discussion - which would come after showing your results. I suggest removing and/or displacing this text. I would also suggest a more traditional organization of the text in Results/Discussion/Conclusions (where two of these three can be grouped together depending on the author's preference). Maybe I am being a bit too rigid here, but when I started reading this section, I was not sure what I was about to read (the introduction-type text at the beginning did not help in this regard).

L151: Please, also list values for the remaining scenarios (if not in the text, in a table including also scaling values for precipitation, Greenland warming, and other relevant values).

L152: Not entirely clear to me what you mean by state-dependence. If it is a state-dependence, shouldn't the value keep increasing for SSP5-8.5? It could be interesting to speculate why this is not the case (e.g., thermodynamic limit at ice surface).

L153: I am not a native speaker, but "analogously" sounds bad.

L150-175: In general, there is a bit of back and forth between historical and projections, which makes the text a bit hard to follow. I would suggest splitting more clearly between historical and projections, or at least avoid this back and forth as much as possible.

L171-L174: "The ensemble mean of the spatially averaged scaling factors agrees with the scaling factors derived from the ensemble mean of the near-surface temperatures" and "...the relationship between the GMT and the spatially averaged seasonal temperatures for some models does not necessarily follow a clear linear relationship.". I find these two sentences somehow contradictory: the first sentence implies that there is linearity ( avg(f(T)) ~ f(avg(T)) ), but the second is not. Am I interpreting this wrong?

L185: Please discuss the historical spatial pattern before projections.

L194-208: I like this analysis of different models; I think it's going to be useful to the community. Perhaps you could try to discuss in some cases why there are differences? E.g., AMOC decline, persistent atmospheric blocking in some models...

L216: See first comment of this section.

L247-261: I find also this part very interesting, and I think the authors could try, again, to link some of the spatial differences to some specific processes (again, I am thinking about AMOC slowdown which would influence moisture transport to southern Greenland). Also, it would be good to frame these results in the context of the baseline precipitation (e.g., present-day northern Greenland quite dry vs south-eastern Greenland more wet).

4. *Modelling the response of the ice sheet*

L287: There is a clear difference between...

L304: Similar comment about AMOC before.

L306: "In fact, the spatial patterns of the precipitation sensitivity agree very well with the ice thickness difference in eastern Greenland observed by the year 2100.". Isn't this obvious as the precipitation sensitivity is derived from the same data you are forcing the ice sheet model with?

L375-L381: I think this part of discussion should be introduced earlier in the text (see comments above about climate processes).

---

## Referee Comment (RC2)

**Review of Bochow et al. "Projections of Precipitation and Temperatures in Greenland and the Impact of Spatially Uniform Anomalies on the Evolution of the Ice Sheet"**

August 5, 2024

**General comments:**

The manuscript by Bochow et al. analyses temperature and precipitation data from a large number of CMIP6 models for the Greenland ice sheet. In their analysis, they find that temperatures over the Greenland ice sheet increase faster than the global mean temperature in the CMIP6 ensemble. Changes in precipitation show a spatially inhomogeneous pattern, raising concerns about the assumption of spatially homogeneous changes made in previous Greenland ice sheet simulations. The authors quantify the effect of this assumption by performing 100,000 year-long simulations under different forcing scenarios using the ice-sheet model PISM.

Overall, I believe the topic is of great interest to the cryospheric community. The manuscript will be a worthwhile contribution to TC. It is in large parts well-written, figures are appropriate, and the drawn conclusions are sound. That said, I have a number of general comments as well as detailed comments listed below that should ideally be addressed. I hope the authors find my comments helpful.

**Specific comments:**

1. The description of the model setup in the methods section is in its current form insufficient. The authors spend >1 page on describing the CMIP6 data, and then dedicate only 1/2 page on the ice-sheet model description. While I certainly appreciate a succinct model description and I am not advocating for repeating every single model equation, I am in favour of self-containing papers. At the moment there is no mention of how the CMIP6 data is fed into the ice-sheet model. What is the temporal resolution of the data and what data is used as input to the dEBM (I presume temperature!). I am also

unclear about what "bootstrapping the model to present-day state" entails. Does that mean parameter values are chosen such that the geometry of the Greenland ice sheet remains close to present day? How do you initialise ice-sheet temperature? How long is the spin up? What is your tuning target (ice velocity, ice geometry or something else)? Moreover, it is mentioned that MAR fields are used in the spin-up procedure. Are anomalies then calculated with respect to the MAR output fields? I presume that the uniform lapse rate and height-induced near-surface temperature change are only used for the spinup, but not for the scenario simulations?

A potential structure to improve this section would be to have a subsection on "PISM model description" followed by "Model spinup", followed by "Scenario experiments" (or similar). To circumvent cluttering the text with parameter values, the authors could also add a parameter and/or experiment table.

2. Related to my previous point, I am surprised that the authors select a model resolution of 20 km. This is a relatively coarse resolution. Given that the authors present a total of 8 simulations over 100,000 years, I do not think that computational demands should be a major constraint. Ideally, the authors redo their simulations with a higher model resolution (10 km) and show that their presented results are still valid. At the very least, the authors should pick one of the higher emission scenarios and show, based on this, that the results are not model resolution dependent.

3. I think the paper could give a more circumspect perspective on the findings as well as potential shortcomings by a more in-depth discussion. I will point them out again in the technical corrections listed below, but a couple of things that I was wondering about as I read the manuscript: "Do you see any differences in the precipitation changes between higher and lower resolution GCM output?". Then, I would like to see some explicit mentioning that the forcing from the CMIP6 models that you use does NOT take into account any changes in ice-sheet geometry.

**Technical corrections:**

Abstract:
L1: I think at some point it would be good to define what you mean by long-term. Depending on the community this could mean anything between centennial to glacial cycle time scales.
L12: Delete "state-of-the-art"

L20: I do not think the projection paper by Edwards et al, 2021 is the best citation for this

L29: I would recommend adding a couple of citations for the stand alone ice-sheet models in addition to the Bochow et al. 2023 study

L30: "to fully Earth System Models (ESMs) with dynamically coupled ice sheets"

L36: Swap order of millennial and deca-millennial

L36-39: What about the PDD method? Isn't that one of the most commonly used parameterisations? Also the citations are extremely PISM "heavy". Maybe worth listing a couple of examples from other ice-sheet models.

L58: Maybe better inaccurate instead of inappropriate

L66-70: Could be worth adding section references to guide the reader better through the manuscript.

L74: Introduce all abbreviations at first mention (SSP was used previously)

L125: I do not understand what "a front-retreat calving based on the observed present-day extent" is? Are you saying that your ice sheet front is not allowed to advance beyond the present-day geometry? Please clarify.

L126: As mentioned above, you need to describe your model setup a lot better including what data you feed from your CMIP6 data into PISM

L126: Again, as suggested above, it would be good to have a experiment overview section or at least a table.

L135: In this section or somewhere in the discussion, please add an explicit mention that your CMIP6 forcing assumes constant present-day ice sheets.

Sections 3.1 and 3.2: Have you had a look if CMIP6 model resolution affects the spatial patterns for the temperature and/or precipitation changes? Especially for precip, I could imagine that higher resolution versions manage to better resolve the topography of Greenland and therefore show a slightly different pattern.

L178: Please add the location of Ilulissatfjord to one of the Figures e.g. Fig 2

L204-205 and L250ff: To me these results are not that surprising. In the high emission scenario you hit your model basically with a sledgehammer that overprints almost all of the internal variability that you might see.

L215-216: Repetition of "similar". Please rephrase.

L246: A bit of a strong claim in my view. Maybe better "fails to capture spatially inhomogeneous patterns". Otherwise, one could argue that using forcing from a climate model that does not account for ice-sheet changes is also not the right choice.

L253, L257: Here and throughout: Delete vague language like "relatively"

L287: replace "to branch" with "to diverge"

L290ff: Here, I started to wonder how much ice dynamics play a role? This could be easily checked by turning off the velocity solve in PISM and keep the initial velocity field. This run would be super fast as you do not solve for the ice dynamics, but would indicate whether ice-sheet dynamics actually matter at all for your simulations.

L337: Is the regrowth due to the GIA feedback that you mentioned earlier?

L369: Can you be more specific about which dynamic processes are involved here? Wind patterns for example?

L397-404: This paragraph comes a little out of the blue. Either delete or improve transition into it.

L418-420: This is a very strong statement. It is also ambiguous. EMICs can easily cover long time scales and are fully coupled. If you are referring to ESMs, you should explicitly mention it. Still, I would rather say "they will remain challenging" or something like "As long as fully coupled ESM simulations on long term timescales are not feasible, we recommend ..."

**Figures:**

The Figures are appropriate and of good quality. For the appendix I suggest that you change the Figure numbering so that for example all precip Figures are contained in a single subsection of the Appendix.

Fig. 1: Dashed lines are somewhat difficult to see. I suggest reducing the transparency. Also consider changing units of subplot b to mm/yr as it is used in the text.

Fig. 2: Here and throughout the manuscript, there is not a coherent use of near-surface temperature, surface temperature, and temperature. I would suggest to chose one and use it consistently.

Fig. 4: Might be worth adding a second y-axis for panel a showing sea-level equivalent. Changes in ice volume in m are more difficult to put into context.

Fig. 5: Same as for Fig. 4. Caption: "... are visible."

Fig. 6: Caption: Please rephrase "huge mistake" as it suggests that one of the simulations is the truth. I would rather say "large discrepancies".

Fig. B1: Consider adjusting colourmap or colourscale. Three of the six panels are basically black.

Fig. D1 and similar: R values and sub-panel titles are not visible. Either increase (I guess model names could also be displayed along the y-axis) or provide a code and a corresponding table where the reader can look up the model name.

Sincerely, Clemens Schannwell

---

## Author Comment (AC1)

**Answers Reviewer 1**

In this paper, the authors provide insights into the importance of considering spatial variations in temperature and precipitation anomalies for accurate modeling of the Greenland ice sheet's response to climate change, highlighting the potential impact of modeling choices on long-term ice sheet stability and sea level rise projections. To do so, Bochow et al. analyzed CMIP6 temperature and precipitation changes in Greenland, deriving spatially resolved and scalar scaling factors for near-surface temperatures and precipitation against global mean temperature and time. The authors then used the Parallel Ice Sheet Model (PISM) to compare the impact of spatially uniform versus spatially resolved anomalies on the Greenland ice sheet's short and long-term behavior. I think this is an interesting paper, worthy of publication for The Cryosphere. I do have though a couple of major comments which I think the authors should address before publication: one regarding how the methodology is presented, and another one regarding an aspect of the methodology (run-time lapse rate and precipitation correction factors).

Thank you for this positive assessment of our manuscript! In the following we will answer all comments and concerns (in blue font).

Major comments:

1) I think that the methodology subsection 2.2 PISM lacks details and is not clear enough. For instance, there are too few details about the initialization procedure and some aspects of the model forcing (calving rates, see specific comments). Also, while I understand in principle how the simulations were forced, I found the text explaining the forcing not entirely clear. For instance, it seems to me that applying spatially variable anomalies of temperature and precipitation is not different from any other study using GCM output (including ISMIP6 studies). In this regard, the link with the scaling factors is not entirely clear (see second comment). I suggest expanding the text explaining clearly the different experiments (perhaps including a table for different experiments, or a diagram showing how the forcing is produced).

We agree that this section was not detailed enough, and we will expand it in a revised version, and we will now also include a table to visualize the different experiments.

2) I was a bit disappointed to read that you applied constant lapse rate and elevation-change-induced precipitation scaling throughout the run. Somehow it is contradicting the premises of the manuscript - that is, going beyond values calculated with previous version of CMIP models. I think that using the values you inferred for precipitation correction would be the missing link between the first part of your results (deriving parameters) and the

second part (ice sheet model simulations). This would shed light not only on scalar vs spatially varying temperature and precipitation, but also on scalar vs spatially varying precipitation correction. If what I suggest is too complicated, it would be enough to test at least different scalar values for the precipitation correction (within the range values found in this study). A similar thing could (should) be done for the lapse rate (although the lapse rate is not calculated in this study: perhaps you could use values taken from Feenstra et al., 2024 "Effect of elevation feedbacks and climate mitigation on future Greenland ice sheet melt", The Cryosphere (preprint).

Thank you for the comment! We think there might be some misunderstanding here. Primarily, we wanted to show the effect of using a scalar/uniform or spatially varying temperature-change-induced, i.e., due to changes of the "background climate" temperature, precipitation correction independent of the elevation-induced effects. Unfortunately, PISM only supports spatially and temporally uniform height-change-induced precipitation correction factors and temporally and spatially uniform lapse rates. That means without restarting the ice sheet model regularly and calculating the precipitation and temperature fields manually, it is not possible to include a spatially varying lapse rate or height-change induced precipitation corrections. We decided to implement the spatially height-change induced precipitation corrections manually and include it now in the revised version alongside the original experiments, at least for the short-term experiments.

What additionally might lead to confusion is that instead of letting PISM internally apply the corrections via the uniform or spatial scaling factors, we directly used the precipitation and temperature anomalies derived from CMIP6 to force the model. In the revised version we force the model directly with the uniform and spatially variable precipitation sensitivities instead of using anomalies to make it more consistent with our analysis.

We also extended our experiments in the revised version and now run all simulations on a 10km resolution instead of 20km. We will now test different scalar height-change induced precipitation correction factors and lapse rates, as much as computational constrains will allow, and additionally run the 85-year simulations with the suggested spatially varying height-change induced precipitation correction factors. Unfortunately, the data from the Feenstra et al. (2024) paper are not published together with the preprint, so we are not able to apply spatially varying lapse rates (easily).

Specific comments:

Abstract

L2: I think uniform anomalies should go first in the sentence than parametrisation schemes.

We changed it accordingly in the revised version.

L3: Is this paper looking at different model parameters, or precipitation correction only? Suggest rephrasing like 'it is often assumed … based on old generation models'.

We rephrased the sentence accordingly to make it clearer. In this manuscript we do not investigate the effect of different model parameters but only the effect of temperature and precipitation anomalies/sensitivities.

L12: I would try to also add here why there is such an overestimation.

We extended the sentence in the revised version.

1. Introduction

L20: surface mass balance is accumulation - ablation; I am not clear if here you also consider ice discharge into the ocean, which seems the case from L23: then I suggest explaining better how mass balance is calculated. Also, missing the reference (probably Otosaka et al. 2023: perhaps you can put together this sentence with the next about global sea level rise).

Thank you for this comment. We agree that we were imprecise with the mass balance and explain it better now.

L25: would explain why (ice margin retreating and losing contact with the ocean).

Thank you, we extended the sentence now.

L27: I would suggest rephrasing the second part, something like: "to ice sheet models coupled to Earth system models of intermediate or full complexity". Or at least you should specify that full complexity ESMs are coupled to a dynamic ice sheet model component - which is not obvious, as most full complexity ESMs are not. Also, I would suggest citing at least ISMIP6 work under the stand-alone ice sheet modelling.

We rephrased the sentence and added the appropriate references.

L31: I find 'latest generation' a bit confusing, is it CMIP5? CESM and UKESM are CMIP6 models with ice sheet coupling. Also, I think you could specifically mention the coupling work with a dynamic Greenland ice sheet component.

We extended the sentence to make clear that we meant the released runs of the ESMs in the CMIP6 intercomparison.

L38: maybe uniform instead of scalar?

Thank you, we changed it accordingly.

L43: I find the notation L(T) a bit confusing, as it looks like a function of the temperature T - but it is instead a constant (although it depends on the temperature). Maybe L_T?

We changed it throughout the manuscript.

L51: is 23C the annual mean temperature between 1996 and 2019? I am not entirely convinced how this simple calculation is relevant for the paper. Perhaps the authors can expand on that?

We wanted to show that already a simple calculation using the Clausius-Clapeyron relationship and Greenland specific values gives a deviation from the commonly used 7-8% for the precipitation sensitivity. The temperature of -23C is the approximate inland annual temperature as defined in the Jiang et al. (2020) paper. We expanded a little bit on it to make clear what we wanted to show with this calculation.

L65: Please add citation for CMIP6.

Thank you, we added a reference.

L66: Please add citation for SSPs. Also, SSP abbreviation was already used before introducing it (L75).

We fixed it in the revised version.

2. Data and Methods

L80: Gaussian with capital G?

Thank you.

L83: What did you do when the land-ice fraction variable was not available?

We only checked the models that have the land-ice fraction variable available. We only do this to verify our approach of using geopandas instead of some model-dependent variable. We agree that ideally one would check the land-ice fractions of all models using some other variable. However, (i) there are simply no land-ice variables available for some models, (ii) we do not see this need here as we are confident that our approach also holds for the other models. We elaborate the sentences now to explain why we look at the land-ice fractions.

L86: precipitations

Thank you, we fixed it.

L99: Perhaps spatially variable scaling factors?

Thank you, "spatially" must have been lost somewhere in the process. We changed it throughout the manuscript where it was missing.

L111: I think this choice should be explained.

This choice was to avoid any noise or outliers on an annual scale in the forcing. We will explain it in the text now.

L117: I think you should give an overview of what has been done here. Also, "mostly" is too vague - please state clearly what has been done differently.

We expanded the section substantially (also see answer major comment #1).

L123: I am a bit confused here. If you are calculating precipitation scaling with temperature, why then use a constant value for the surface-height-induced near-surface temperature change? I am not necessarily saying you should not do that, but I think you should explain this choice.

Thank you for this comment, it is indeed not obvious from the text. We would have liked to use spatially varying values for the surface-height-induced near-surface temperature changes and precipitation changes, but PISM does not offer this feature (yet). However, we decided to manually implement spatially varying factors in PISM. We will at least run the short simulations with this feature in the revised manuscript.

L125: I suggest adding a few more details about front-retreat calving. It's not clear to me how and why this is implemented, especially for the long-term runs.

Thanks, we extended the model description section and now give more details overall, including the calving implementation.

L127: I assume what comes between L127 and L132 is about the spin-up. I suggest then moving the last sentence (resolution) upwards, as I imagine it also regards the future projections.

Thank you!

3. Greenlands climate in CMIP6

L135-L140: the beginning of this section feels a bit strange, as the first lines seem to belong either to the introduction or the discussion - which would come after showing your results. I suggest removing and/or displacing this text. I would also suggest a more traditional organization of the text in Results/Discussion/Conclusions (where two of these three can be grouped together depending on the author's preference). Maybe I am being a bit too rigid here, but when I started reading this section, I was not sure what I was about to read (the introduction-type text at the beginning did not help in this regard).

We will try to reorganize the manuscript in the revised version and try to follow a more classical order of the sections.

L151: Please, also list values for the remaining scenarios (if not in the text, in a table including also scaling values for precipitation, Greenland warming, and other relevant values).

Thank you for the suggestion, we now mention all important values in the text, not only in the figures.

L152: Not entirely clear to me what you mean by state-dependence. If it is a state-dependence, shouldn't the value keep increasing for SSP5-8.5? It could be interesting to speculate why this is not the case (e.g., thermodynamic limit at ice surface).

Here, state dependence does not necessarily mean a monotonous increase with increasing temperature. But there is an obvious dependence of the precipitation sensitivity on the emission scenario which is increasing from SSP1-2.6 to SSP5-8.5 (Fig. 3a). When looking at Fig. 3a in the SSP5-8.5 scenario, there also seems to be a slightly stronger increase in the precipitation rates at higher temperatures than for lower temperatures. We will try to make this clearer in the revised version.

L153: I am not a native speaker, but "analogously" sounds bad.

Thanks for noticing, we rephrased the sentence.

L150-175: In general, there is a bit of back and forth between historical and projections, which makes the text a bit hard to follow. I would suggest splitting more clearly between historical and projections, or at least avoid this back and forth as much as possible.

We tried to follow a different organization of the results that doesn't necessarily correspond to the timeline of the scenarios. However, we agree that there is too much back and forth between historical and future scenarios. In the revised version we try to avoid this as much as possible.

L171-L174: "The ensemble mean of the spatially averaged scaling factors agrees with the scaling factors derived from the ensemble mean of the near-surface temperatures" and "...the relationship between the GMT and the spatially averaged seasonal temperatures for some models does not necessarily follow a clear linear relationship.". I find these two sentences somehow contradictory: the first sentence implies that there is linearity (avg(f(T)) ~ f(avg(T)) ), but the second is not. Am I interpreting this wrong?

Thank you for the comment. Here we indeed mean (avg(f(T)) ~ f(avg(T))) with avg being the mean over the ensemble but that does not necessarily imply that f(T) for each member has to be linear but approximately linear since we do not state (avg(f(T)) = f(avg(T))). We extend these sentences to resolve this lack of clarity.

L185: Please discuss the historical spatial pattern before projections.

Thanks for this suggestion, we will change it in the revised version.

L194-208: I like this analysis of different models; I think it's going to be useful to the community. Perhaps you could try to discuss in some cases why there are differences? E.g., AMOC decline, persistent atmospheric blocking in some models…

In the revised version, we will discuss in more detail potential reasons for the different model responses.

L216: See first comment of this section.

Thanks.

L247-261: I find also this part very interesting, and I think the authors could try, again, to link some of the spatial differences to some specific processes (again, I am thinking about AMOC slowdown which would influence moisture transport to southern Greenland). Also, it would be good to frame these results in the context of the baseline precipitation (e.g., present-day northern Greenland quite dry vs south-eastern Greenland more wet).

Thanks for this suggestion, we now discuss possible reasons for the observed difference in the model responses in the revised manuscript. We thank the reviewer for marking this link between the AMOC response and the climate in Greenland. Indeed, we find a relationship between the decline in the AMOC strength and the precipitation rates/temperature anomalies and discuss them in the revised manuscript.

4. Modelling the response of the ice sheet

L287: There is a clear difference between…

Thanks, we change it in the revised version.

L304: Similar comment about AMOC before.

Thanks, see other comments.

L306: "In fact, the spatial patterns of the precipitation sensitivity agree very well with the ice thickness difference in eastern Greenland observed by the year 2100.". Isn't this obvious as the precipitation sensitivity is derived from the same data you are forcing the ice sheet model with?

We agree that this is more or less obvious. The order of the sentence was probably not optimal, we changed it in the revised version.

L375-L381: I think this part of discussion should be introduced earlier in the text (see comments above about climate processes).

Thanks for the suggestion, we will move this part of the discussion in the revised manuscript.

---

## Author Comment (AC2)

**Answers Reviewer 2**

General comments:

The manuscript by Bochow et al. analyses temperature and precipitation data from a large number of CMIP6 models for the Greenland ice sheet. In their analysis, they find that temperatures over the Greenland ice sheet increase faster than the global mean temperature in the CMIP6 ensemble. Changes in precipitation show a spatially inhomogeneous pattern, raising concerns about the assumption of spatially homogeneous changes made in previous Greenland ice sheet simulations. The authors quantify the effect of this assumption by performing 100,000 year-long simulations under different forcing scenarios using the ice-sheet model PISM. Overall, I believe the topic is of great interest to the cryospheric community. The manuscript will be a worthwhile contribution to TC. It is in large parts well-written, figures are appropriate and the drawn conclusions are sound. That said, I have a number of general comments as well as detailed comments listed below that should ideally be addressed. I hope the authors find my comments helpful.

Thank you for your positive assessment of our manuscript. Below, we address all of your comments and concerns (blue font).

Specific comments:

1. The description of the model setup in the methods section is in its current form insufficient. The authors spend >1 page on describing the CMIP6 data, and then dedicate only 1/2 page on the ice-sheet model description. While I certainly appreciate a succinct model description and I am not advocating for repeating every single model equation, I am in favour of self-containing papers. At the moment there is no mention of how the CMIP6 data is fed into the ice-sheet model. What is the temporal resolution of the data and what data is used as input to the dEBM (I presume temperature!). I am also unclear about what "bootstrapping the model to present-day state" entails. Does that mean parameter values are chosen such that the geometry of the Greenland ice sheet remains close to present day? How do you initialise ice-sheet temperature? How long is the spin up? What is your tuning target (ice velocity, ice geometry or something else)? Moreover, it is mentioned that MAR fields are used in the spin-up procedure. Are anomalies then calculated with respect to the MAR output fields? I presume that the uniform lapse rate and height-induced near-surface temperature change are only used for the spinup, but not for the scenario simulations?

A potential structure to improve this section would be to have a subsection on "PISM model description" followed by "Model spinup", followed by "Scenario experiments" (or similar).

To circumvent cluttering the text with parameter values, the authors could also add a parameter and/or experiment table.

Thank you for your comment. We agree that the model description section was insufficient. Based on your feedback and that of the other reviewer, we have expanded the section to address all your concerns and questions. As suggested, we have divided the section into "Model Description," "Spinup," and "Experiments." Additionally, we have included a table summarizing the different experiment setups in the revised version.

2. Related to my previous point, I am surprised that the authors select a model resolution of 20 km. This is a relatively coarse resolution. Given that the authors present a total of 8 simulations over 100,000 years, I do not think that computational demands should be a major constraint. Ideally, the authors redo their simulations with a higher model resolution (10 km) and show that their presented results are still valid. At the very least, the authors should pick one of the higher emission scenarios and show, based on this, that the results are not model resolution dependent.

Thank you for the comment! In the initial manuscript we did not want to put too much weight on the model simulation part. We rather wanted to show the effect of the uniform vs spatially variable sensitivities by running idealized experiments. However, we now considerably extended all our simulations by varying some important parameters and run everything on a 10km resolution.

3. I think the paper could give a more circumspect perspective on the findings as well as potential shortcomings by a more in-depth discussion. I will point them out again in the technical corrections listed below, but a couple of things that I was wondering about as I read the manuscript: "Do you see any differences in the precipitation changes between higher and lower resolution GCM output?". Then, I would like to see some explicit mentioning that the forcing from the CMIP6 models that you use does NOT take into account any changes in ice-sheet geometry.

We extend the discussion in the revised manuscript and discuss, among others, the potential differences between higher and lower resolution models and the influence of different AMOC responses across the models on the climate in Greenland.

Technical corrections:

Abstract:

L1: I think at some point it would be good to define what you mean by long-term. Depending on the community this could mean anything between centennial to glacial cycle time scales.

We define it now in the abstract as "millennial time scale and beyond".

L12: Delete "state-of-the-art"

Thanks.

L20: I do not think the projection paper by Edwards et al, 2021 is the best citation for this

Indeed, we now cite the primary source for this number.

L29: I would recommend adding a couple of citations for the stand alone ice-sheet models in addition to the Bochow et al. 2023 study

Thank you, we added two more references to this part of the sentence.

L30: "to fully Earth System Models (ESMs) with dynamically coupled ice sheets"

We rephrased the sentence.

L36: Swap order of millennial and deca-millennial

Thanks for the comment; however, we don't see a reason to change the word order, as we progress from a shorter (millennial) to a longer time scale (deca-millennial).

L36-39: What about the PDD method? Isn't that one of the most commonly used parameterisations? Also the citations are extremely PISM "heavy". Maybe worth listing a couple of examples from other ice-sheet models.

Thank you. Yes, PDD is probably one of the most used parameterization methods for long-term ice sheet modelling. However, in this paper we concentrate on the assumptions made for the climate fields that are fed into these SMB parameterization methods rather than the validity of the SMB parameterization methods themselves. We rephrased the sentences to make it clearer and added a few more non-PISM references.

L58: Maybe better inaccurate instead of inappropriate

Thank you, we changed the word.

L66-70: Could be worth adding section references to guide the reader better through the manuscript.

Thanks for this suggestion.

L74: Introduce all abbreviations at first mention (SSP was used previously)

We fixed that in the revised manuscript.

L125: I do not understand what "a front-retreat calving based on the observed present-day extent" is? Are you saying that your ice sheet front is not allowed to advance beyond the present-day geometry? Please clarify.

Yes, that is what it means. We clarify it now in the method section.

L126: As mentioned above, you need to describe your model setup a lot better including what data you feed from your CMIP6 data into PISM

We hope our revised method section is clearer now.

L126: Again, as suggested above, it would be good to have a experiment overview section or at least a table.

Thanks for this suggestion, we now have a table describing the different experiments.

L135: In this section or somewhere in the discussion, please add an explicit mention that your CMIP6 forcing assumes constant present-day ice sheets. Sections 3.1 and 3.2: Have you had a look if CMIP6 model resolution affects the spatial patterns for the temperature and/or precipitation changes? Especially for precip, I could imagine that higher resolution versions manage to better resolve the topography of Greenland and therefore show a slightly different pattern.

We mention now explicitly that the CMIP6 data assumes a fixed ice sheet topography. We will also discuss the model resolution, among other things, as a reason for differences in the spatial patterns. However, we also want to note that we think a thorough investigation of possible reasons for different model responses might be beyond this paper's scope.

L178: Please add the location of Ilulissatfjord to one of the Figures e.g. Fig 2

Thank you for this suggestion, we depict Ilulissatfjord now in Fig. 2.

L204-205 and L250ff: To me these results are not that surprising. In the high emission scenario you hit your model basically with a sledgehammer that overprints almost all of the internal variability that you might see.

Agreed. We removed the word "interestingly" in L250 as it is not very surprising.

L215-216: Repetition of "similar". Please rephrase.

Thanks for pointing this out.

L246: A bit of a strong claim in my view. Maybe better "fails to capture spatially inhomogeneous patterns". Otherwise, one could argue that using forcing from a climate model that does not account for ice-sheet changes is also not the right choice.

We agree and change the sentence accordingly.

L253, L257: Here and throughout: Delete vague language like "relatively"

We reduced the word "relatively" throughout the manuscript.

L287: replace "to branch" with "to diverge"

We changed the wording accordingly.

L290ff: Here, I started to wonder how much ice dynamics play a role? This could be easily checked by turning off the velocity solve in PISM and keep the initial velocity field. This run would be super fast as you do not solve for the ice dynamics, but would indicate whether ice-sheet dynamics actually matter at all for your simulations.

Thanks, we now include a short paragraph about the dynamic contribution to the ice thickness change and also include a new figure.

L337: Is the regrowth due to the GIA feedback that you mentioned earlier?

Yes, we mention it now.

L369: Can you be more specific about which dynamic processes are involved here? Wind patterns for example?

Thanks, we are more specific now.

L397-404: This paragraph comes a little out of the blue. Either delete or improve transition into it.

Agreed, we moved it further up.

L418-420: This is a very strong statement. It is also ambiguous. EMICs can easily cover long time scales and are fully coupled. If you are referring to ESMs, you should explicitly mention it. Still, I would rather say "they will remain challenging" or something like "As long as fully coupled ESM simulations on long term timescales are not feasible, we recommend ..."

Indeed, we meant fully coupled ESM simulations, we mention it now and rephrased the sentence according to your suggestion.

Figures:

The Figures are appropriate and of good quality. For the appendix I suggest that you change the Figure numbering so that for example all precip Figures are contained in a single subsection of the Appendix.

Fig. 1: Dashed lines are somewhat difficult to see. I suggest reducing the transparency. Also consider changing units of subplot b to mm/yr as it is used in the text.

Fig. 2: Here and throughout the manuscript, there is not a coherent use of near-surface temperature, surface temperature, and temperature. I would suggest to chose one and use it consistently.

Fig. 4: Might be worth adding a second y-axis for panel a showing sea-level equivalent. Changes in ice volume in m are more difficult to put into context.

Fig. 5: Same as for Fig. 4. Caption: "... are visible."

Fig. 6: Caption: Please rephrase "huge mistake" as it suggests that one of the simulations is the truth. I would rather say "large discrepancies".

Fig. B1: Consider adjusting colourmap or colourscale. Three of the six panels are basically black.

Fig. D1 and similar: R values and sub-panel titles are not visible. Either increase (I guess model names could also be displayed along the y-axis) or provide a code and a corresponding table where the reader can look up the model name.

Sincerely, Clemens Schannwell

Thank you for the comments on the figures! We will incorporate all your suggestions.

---

## Author Response (AR2)

**Answers Reviewer**

Overall, I am happy with the revisions made by the authors and I believe the manuscript is almost ready for publication. I only have a number of smaller comments that I think should be addressed before publication:

Thank you for this positive assessment of our revision! Below, we answer any remaining concerns in blue font.

Abstract: L1-5 There is a lot of "assume" and "often" in the first few sentences. I suggest to rework these first few sentences and try avoiding unnecessary repetitions.

Thanks, we agree and rephrased the abstract slightly.

L8: Why do you not give the full range but instead write "less than 0%"?

We do now.

L12: introduce abbreviation "GMT" at first mention

Thank you, we fixed it.

L15: Please mention at least one of the "key regions". Could be done with by adding "such as…"

Thank you, we clarify it now.

L37-38: I think it would be appropriate to add at least one citation of an earlier effort to include dynamic ice sheets into ESMs (not EMICS). For example Mikolajewicz et al. 2007, GRL, doi: 10.1029/2007GL031173 or Vizcaíno et al. 2008, Clim Dyn., doi: 10.1007/s00382-008-0369-7

Thank you, we include these references now.

L39-40: I am a bit biased, but I suggest to change "make it currently virtually impossible" to "currently challenging" or similar. As our group's paper shows, simulations of the entire last deglaciation (26,000 years) are certainly feasible now (see doi.org/10.5194/cp-2024-55).

We rephrased the sentence.

L74: change "not necessarily true" to "unlikely to be true"

We rephrased the sentence accordingly.

L92: I did find the last section of the sentence confusing and hence suggest to delete it. If the authors would like to keep the sentence, then I suggest at least that they add that these numbers refer to the radiative forcing and that the numbers following the hyphen indicate the increase in watts per meter squared.
We agree and deleted the last part of the sentence.

L94 Are "tas" and "pr" really needed? It is not used anywhere else right?

We included these variables to allow for better reproducibility since the nomenclature of variables in the CMIP6 ensemble is not very clear. Therefore, we decided to leave it like it is.

L140: parentheses missing around Zeitz et al. (2021).

Thank you, we fixed it.

L161: "We spin up" instead of "We spinup"

Thank you, we fixed it.

L211-212: I do not understand why you need to normalise the ice volume? Do your simulations not start from the same initial state anyway?

We agree that it is not necessary in this case, especially since we do not give any prognostic sea-level estimates. We barely followed the methods from other papers to give a better picture of the expected sea-level contribution since our initial states have of course not the same ice volume as the observed volume of the GrIS. Since this decision to normalize the ice volume does not have any impact on the results, we decided to leave it as it is.

L264-L265 "almost the entire Greenland Ice Sheet shows..."

Fixed.

L276: Here and throughout, change "similar pictures arise" to "similar patterns are observed" or similar.

Thank you for this suggestion. We changed it accordingly.

L287: "magnitudes of the temperatures"

Fixed.

L307 "has a moderate correlation"

Thank you, we rephrased it accordingly.

L341-343 Do you mean in comparison to the historical period? Because all of the emission scenarios should be the same for the historical period, right?

Unfortunately, it is not clear to us what you refer to. We give the precipitation sensitivity for each emission scenario over the respective time period. That is, for the future emission scenarios for the 21$^{st}$ century and for the historical scenario for the historical period (1850-present). We put the sensitivity for the historical period first now to make it more consistent with the temporal order of the scenarios.

L431: Delete "Little surprisingly"

Fixed.

Section 4: I think it is worth adding a few sentences/paragraphs that spatially varying precip. sensitivites (exp iii) do not seem to matter in your simulations.

We agree and added two sentences.

L604: Again, I think this is too strong a statement. Such simulations exist, see comment above. My suggestion would be something like: "remain rare" or similar.

We rephrased the sentence.

Figures:

Fig.4 Second -yaxis. I find the term "sea-level rise" not precise enough. I think something like "Ice volume [m of sea-level equivalent]"would be more appropriate.

We changed it throughout all relevant figures.

Fig. 3 caption: Please add that the contour is white and make the contour thicker and maybe even label the contour like you do in Fig. 2

We changed the caption and figure accordingly.

Fig. 5: Can you increase the legend fontsize?

Done.

Fig. 6 caption: Please add somewhere that dashed-dotted lines are on top of the dashed lines to the caption

Thanks for this suggestion, we changed the caption accordingly.